# A *Panax quinquefolius*-Based Preparation Prevents the Impact of 5-FU on Activity/Exploration Behaviors and Not on Cognitive Functions Mitigating Gut Microbiota and Inflammation in Mice

**DOI:** 10.3390/cancers14184403

**Published:** 2022-09-10

**Authors:** Renaud Parment, Martine Dubois, Laurence Desrues, Alexandre Mutel, Kléouforo-Paul Dembélé, Nicolas Belin, Laure Tron, Charlène Guérin, Moïse Coëffier, Vincent Compère, Céline Féger, Florence Joly, Pascal Hilber, David Ribet, Hélène Castel

**Affiliations:** 1Normandie University, UNIROUEN, INSERM, U1245, Cancer and Brain Genomics, 76000 Rouen, France; 2Institute for Research and Innovation in Biomedicine (IRIB), 76000 Rouen, France; 3Cancer and Cognition Platform, Ligue Nationale Contre le Cancer, 14000 Caen, France; 4Les Laboratoires Phytodia, 67412 Illkirch, France; 5Clinical Research Department, Centre François Baclesse, 14000 Caen, France; 6Normandie University, UNICAEN, INSERM, ANTICIPE, 14000 Caen, France; 7Normandie University, UNIROUEN, INSERM UMR 1073, Nutrition, Inflammation et axe Microbiote-Intestin-Cerveau, 76000 Rouen, France; 8CHU Rouen, Department of Nutrition, 76000 Rouen, France; 9CHU Rouen, Department of Anesthesia and Critical Care, 76000 Rouen, France; 10Olisma, 23110 Evaux-les-Bains, France; 11CHU Caen, 14000 Caen, France

**Keywords:** chemotherapy, behavioral mouse model, *Panax quinquefolius*-based solution, activity and exploration, cognitive functions, IL-6, gut microbiota, intestinal inflammation

## Abstract

**Simple Summary:**

Chemotherapy-related cognitive impairment (CRCI) and fatigue worsen the quality of life (QoL) of cancer patients. Multicenter studies have shown that *Panax quinquefolius* and vitamin C, respectively, were effective in reducing the symptoms of fatigue in treated cancer patients. We developed a behavioral C57Bl/6j mouse model to study the impact of 5-Fluorouracil (5-FU) chemotherapy on activity/fatigue, emotional reactivity and cognitive functions. We used this model to evaluate the potentially beneficial role of a *Panax quinquefolius*-based solution containing vitamin C (Qiseng^®^) or vitamin C alone in these chemotherapy side effects. We established that Qiseng^®^ prevents the reduction in activity/exploration and symptoms of fatigue induced by 5-FU and dampens chemotherapy-induced intestinal dysbiosis and systemic inflammation. We further showed that Qiseng^®^ decreases macrophage infiltration in the intestinal compartment, thus preventing, at least in part, the systemic elevation of IL-6 and MCP-1 and further reducing the neuroinflammation likely responsible for the fatigue induced by chemotherapy, a major advance toward improving the QoL of patients.

**Abstract:**

Chemotherapy-related cognitive impairment (CRCI) and fatigue constitute common complaints among cancer patient survivors. *Panax quinquefolius* has been shown to be effective against fatigue in treated cancer patients. We developed a behavioral C57Bl/6j mouse model to study the role of a *Panax quinquefolius*-based solution containing vitamin C (Qiseng^®^) or vitamin C alone in activity/fatigue, emotional reactivity and cognitive functions impacted by 5-Fluorouracil (5-FU) chemotherapy. 5-FU significantly reduces the locomotor/exploration activity potentially associated with fatigue, evokes spatial cognitive impairments and leads to a decreased neurogenesis within the hippocampus (Hp). Qiseng^®^ fully prevents the impact of chemotherapy on activity/fatigue and on neurogenesis, specifically in the ventral Hp. We observed that the chemotherapy treatment induces intestinal damage and inflammation associated with increased levels of Lactobacilli in mouse gut microbiota and increased expression of plasma pro-inflammatory cytokines, notably IL-6 and MCP-1. We demonstrated that Qiseng^®^ prevents the 5-FU-induced increase in Lactobacilli levels and further compensates the 5-FU-induced cytokine release. Concomitantly, in the brains of 5-FU-treated mice, Qiseng^®^ partially attenuates the IL-6 receptor gp130 expression associated with a decreased proliferation of neural stem cells in the Hp. In conclusion, Qiseng^®^ prevents the symptoms of fatigue, reduced chemotherapy-induced neuroinflammation and altered neurogenesis, while regulating the mouse gut microbiota composition, thus protecting against intestinal and systemic inflammation.

## 1. Introduction

Chemotherapy-related cognitive impairment (CRCI) may occur during or after chemotherapy and constitutes one of the main concerns among cancer patient survivors. Among the reported CRCI, deficits in memory, concentration, attention and executive functions can be detected in 20–75% of patients during the course of chemotherapy and up to 1–2 years after its completion in a smaller proportion of patients [1]. These impacted cognitive domains should also be considered according to other potential disorders, including anxiety/depression states, sleep disturbances and fatigue [2,3], which drastically worsen the quality of life (QoL) of cancer patients. These non-cognitive disorders may also be directly related to cancer treatments and can interact with the CRCI, possibly through distinct and common neurobiological mechanisms.

The study of brain activity measured by fMRI has previously indicated a more extensive activation of the neural circuits involved in the working memory process, suggesting chemotherapy-induced selective damages within the prefrontal cortex, basal ganglia and/or cerebellum, which may persist for a few months to years after chemotherapy completion [4,5]. In addition, structural alterations, such as decreased volume of the gray and white matters in the prefrontal cortex and parahippocampal gyrus, have been observed in patients after an episode of cancer [3]. These observations can be, at least in part, supported by a recent study showing that Olig2-oligodendrocyte precursors were shown depleted in the subcortical white matter, while the pool remained relatively preserved in the gray matter [6] of post-mortem frontal lobe tissues of young adults treated with chemotherapy.

For many years, the neurobiological mechanisms that underpin CRCI, mood disorder and fatigue in cancer patients remained a clinical dark box, even if some biological fluid analyses pointed out an association between plasma cytokines, such as IL-1β, IL-6, TNF-α, TNF-RII, and cognitive dysfunctions or fatigue [7]. In patients with various cancers in whom the prevalence of depressive and anxiety symptoms were around 23 and 19%, respectively [8], there is, however, a lack of association between cytokine levels and mood, most of them receiving at least chemotherapy. Of note, some meta-analyses established that cytokines, including IL-6, IL-10, TNF-α and MCP-1, were found increased in patients with depression compared with healthy controls [9,10]. In addition to cytokines, selective lipids in the cerebral spinal fluids (CRF) or genetic polymorphisms, such as APOE-4 and COMT-val, have been associated with cognitive disorders and CRCI [7]. More recently, in breast cancer survivors, markers of cellular aging were prematurely detected, and this was interpreted as the long-lasting deleterious effects of chemotherapy [11]. Breast cancer patients receiving chemotherapy and/or radiotherapy were more likely to exhibit higher levels of DNA damage and lower telomerase activity 3 to 6 years after the end of the treatment. Such alterations were independently associated with inflammatory reactions revealed by the release of plasma TNF-RII [11]. However, it remains difficult to bridge the gap between these chemotherapy-associated systemic consequences and cognition, fatigue or mood disorders [12,13]. Various hormonal and inflammatory pathways, oxidative stress and/or blood–brain barrier (BBB) permeability were also thought to be linked with altered cognitive performance in the context of cancer. Interestingly, IL-6, although increased in plasma levels in the presence of cancer, is not significantly associated with cognitive disorders [7] but correlated with fatigue [14]. Similarly, it has been shown in C57Bl6 mice that a single intraperitoneal injection of IL-6 produced a significant decrease in open-field locomotor activity [15], potentially associated with fatigue in the animal models. In rodents, the exposure to chemotherapy triggers oxidative stress and increased expression of the nuclear factor kappa-B (NF-κB), which leads to pro-inflammatory cytokine secretion, such as tumor necrosis factor-α (TNF-α), IL-1β and IL-6, and then to tissue damage and apoptosis [16,17]. The co-administration of antioxidants may prevent oxidative damage induced by chemotherapy and the associated cerebral consequences, cognitive dysfunctions but also fatigue.

Several studies have shown a correlation between fatigue and chemotherapy. Around 82–96% of patients undergoing chemotherapy suffer from fatigue during their treatment [18]. Understanding and preventing this noticeable effect is therefore essential for cancer patients and their QoL. A multicenter study cohort of 125 breast cancer patients with standard cancer treatment (surgery, chemotherapy, radiation and hormonal therapy) demonstrated a significant fatigue improvement in patients receiving intravenous vitamin C (7.5 g per week for 4 weeks) compared with the control group, during and 6 months after this treatment [19]. In addition, an American multicenter randomized controlled study of 364 cancer patients during or after anti-cancer treatment showed a significant reduction in fatigue in patients treated for 8 weeks with 2 g/day of *Panax quinquefolius*. This American ginseng-based treatment displays powerful antioxidant activities, at least via bioactive-derived compounds, ginsenosides [20]. These bioactive compounds have both stimulatory and inhibitory effects on the central nervous system [21,22], modulate neurotransmission, favor memory and learning and promote neuroprotection, notably by the action of the ginsenosides Rb1 [23] and Rd [24]. Accordingly, long-term ginsenoside consumption by aged mice prevented impaired learning and memory during the Morris water maze (MWM) test by decreasing oxidative stress, as measured in serum, and by up-regulating the plasticity-related proteins PSD95 and BDNF in the Hp [25]. In oncology, daily oral administration of ginseng or ginsenoside Rb1 in mice bearing colon C26 cancer cells results in reduced levels of TNF-α and IL-6 cytokines in serum, without modifications of tumor growth [26]. Additionally, in rats, ginseng extracts can partially restore the learning and spatial memory deficits induced by cisplatin and reduce neuroinflammation and oxidative stress in the Hp [27]. 

Together, these studies indicate that *Panax quinquefolius* may be beneficial by preventing the inflammatory and neurotoxic effects of chemotherapy to preserve the cognitive functions and/or potential mood disorders or fatigue encountered during CRCI. The aim of the study was first to evaluate the role of 5-FU chemotherapy in locomotion/exploration, the nychthemeral cycle associated with fatigue, emotional reactivity and cognitive functions in young adult mice and then to determine the potentially beneficial impact of a *Panax quinquefolius*-based solution containing vitamin C (Qiseng^®^), or vitamin C alone, on chemotherapy-induced gut microbiota dysbiosis and inflammation, systemic inflammation and hippocampal neurobiological defects associated with behavioral alterations.

## 2. Materials and Methods

### 2.1. Animals

The experiments were conducted on male C57Bl/6J Rj mice obtained from Janvier (Le Genest Saint Isle, France) at the age of 7 weeks and divided into different groups. Mice were then in-housed (animal facility of the University of Rouen Normandie) under controlled standard environmental conditions: 22+/1 °C, 8–10 animals per box with enrichment (lignocel^®^, JRS, Rosenberg, Germany), under an inverse light/dark cycle of 12/12 h (light on: 00–12 a.m.), with water and food available *ad libitum*. After an adaptation period of 2 weeks, the animals were handled and weighted daily (familiarization) until the end of the experimental period. The administration of treatments began when mice were 9 weeks old. The number and the suffering of animals were minimized in accordance with the directives of the European Parliament and Council Directive (2010/63/EU) and the Council for the Protection of Animals Used for Scientific Purposes. The project was approved by the regional “*Comité d’Ethique Normandie en Matière d’Expérimentation Animale*” and the French Ministry of Higher Education, Research and Innovation under the project number: “#20779-201902001322662 v7”. The experiments were carried out under the supervision of authorized investigators (H.C. and M.D.).

### 2.2. Treatments and Experimental Procedures

The chemotherapy 5-FU (60 mg/kg, Sigma-Aldrich, F8423-5G, St. Quentin, France) or the NaCl saline solution (0.9%) was administered intraperitoneally (i.p.) once a week for 3 weeks (W1, W2 and W3) using a 26-gauge Terumo needle. A formulation composed of American ginseng (*Panax quinquefolius*, *P. qfolius*, 140 mg/kg) supplemented with vitamin C (19 mg/kg, as stabilizer) and containing 15% of ginsenosides was referred to as the Qiseng^®^ solution (Natsuca, Castres, France). The placebo solution, excluding *P. qfolius* and vitamin C, contained all ingredients of the formulated product, e.g., Vegetable glycerin, Stevia sweetener, Raspberry flavor, and was provided by Natsuca. Vitamin C (19 mg/kg/day, Sigma-Aldrich, 95209-50G) was diluted in the placebo. Group-housed mice were administered with 100 µL/10 g 5 times a week for 2 (W2 and W3, Sub-studies 2–4) or 3 weeks (W2, W3, W4, Sub-study 1) depending on the experimental procedure. Treatments were given *per os* using a 22-gauge feeding needle (Fine Science Tools, Heidelberg, Germany). During treatments, the animals were weighted daily. All treatments were administered prior to the active phase of mice (9–12 am), and behavioral tests were carried out during this phase (from 1 pm). In one experimental sub-study1 (*n* = 12–14 animals), 4 i.p. injections of 5′-bromo-2′-deoxyuridine (BrdU, 50 mg/kg) (Sigma-Aldrich, B5002-5G) were administered over a period of 7 days to label the brain neural progenitor cells and allow a study of neurogenesis at the end of the behavioral test period from the collected brains. In Sub-studis 2–4, 4 i.p. injections of BrdU (50 mg/kg) were administered over a period of 7 days to label the brain neural progenitor cells and allow a study of proliferation from the collected brains 4 h after the last injection (Sub-study 2). Serum and cecum were sampled in Sub-study 3, and plasma, intestine and brains were sampled in Sub-study 4.

### 2.3. Behavioral Assessments

Emotional reactivity and cognitive functions were assessed in Sub-study 1 during the 3-week post-treatment period. Spontaneous activity and exploration were evaluated by using the open field test (OFT) and the light/dark box (LDB); anxiety and depressive-like behaviors were investigated by means of the elevated plus maze (EPM) and the tail suspension (TST) and forced swimming (FST) tests, respectively. Spatial learning and memory as well as behavioral flexibility were measured with the MWMT. All apparatus, except MWM, were systematically cleaned with 70% alcohol whenever an animal is placed in the device.

### 2.4. Open Field Test—OFT

Mouse activity and exploration were measured in an open-field apparatus 45 × 45 × 31 cm for 10 min. At the beginning of the test, mice were placed in the center of the device while they tendto walk in the periphery along the walls. Since the distance traveled and time spent in the center are an indication of anxiety [28,29], the distance traveled, the number of rearings and the duration of grooming episodes were also recorded for activity during 10 min in the various zones of the device using a video-tracking camera-based software anymaze (Anymaze, Stoelting, Dublin, Ireland). 

In order to evaluate the locomotor activity in a closed and unknown environment, we used the OFT 2 days after the last oral gavage with the placebo, vitamin C or Qiseng^®^ and 10 days after the last injection of NaCl or 5-FU.

### 2.5. Elevated plus Maze—EPM

The anxiety-like behaviors of mice were assessed by means of the EPM [30]. The device consists of four arms: two opposite secure arms closed by 19.5 cm vertical walls and two aversive open arms perpendicular to the previous ones with a 0.5 cm high rim (each arm is 25 cm long and 5 cm wide) and a central area of 5 × 5 cm, elevated at 41.5 cm above the ground. At the beginning of the test, the mouse was placed in the central area of the maze with its head pointing toward an open arm. The times spent in the open and closed arms were measured. The animal’s behavior was videotaped and analyzed using the Anymaze software (Stoelting, Dublin, Ireland) for a single 5 min session.

### 2.6. Light/Dark Box—LDB

Animal impulsive behavior was assessed in the LDB test [31]. The device comprised two compartments: a “dark” area (40 × 19 cm) with dense black walls (height 35 cm) and a lid, and a “light” area (20.3 × 40 cm) with transparent walls with illumination of 85 lux. The two compartments were separated by a wall with a hole in the lower center (7 × 7 cm). At the beginning of the test, the animal was placed in the dark compartment with familiar sawdust sprinkled on the bottom of the dark compartment. The risk assessment was evaluated by counting the number of attempts to enter, with the head and front legs extending into the lighted area and the hind legs remaining in the dark compartment. The time taken to exit into the lighted compartment (4-legged criterion), the number of entries and the time spent in the lighted compartment, as well as the number of wall-leaning in the lighted compartment, were measured for 10 min.

### 2.7. Tail Suspension Test—TST and Forced Swim Test—FST

Depressive-like behavior was assessed by the TST and the FST, which are standard procedures commonly used to evaluate the antidepressant activity of pharmacological compounds [32,33]. In TST, mice were suspended by their tails from a hook using Scotch^®^ Tape (3M France, Cergy-Pontoise, France) at 20 cm above the floor and surrounded by an enclosure (40 cm high, 30 cm wide and 30 cm deep). The total duration of immobility (passive hanging) between periods of agitation to avoid an aversive situation was measured over a period of 6 min. The latency at the first immobility was also analyzed. In FST, mice were placed for 6 min in a cylinder (17.5 cm diameter) filled with tap water at 25 °C with no possibility of escaping. The duration of immobility (excluding movements necessary to keep the head out of the water or to float) was measured as an indication of behavioral despair. The latency at first immobility was also measured.

### 2.8. Morris Water Maze Test—MWMT

Spatial learning and memory performances and learning flexibility were evaluated using the MWM. Mice placed in a water tank had to escape from the water by finding a hidden platform (9.7 cm in diameter) placed in the pool. The pool consisted of a cylindrical basin (100 cm diameter) filled with water to a height of 40 cm, maintained at 22 ± 1 °C, made opaque by a white inert, non-toxic aqueous acrylic emulsion (Accusol OP 301, Viewpoint^®^, Lyon, France). The aquatic device was virtually divided into 4 quadrants: North-West (NW), North-East (NE), South-East (SE) and South-West (SW). Animal behavior and swim trajectories were analyzed with Anymaze. The tests took place over several sessions, as follows. On the first day, the animals were familiarized with the device, and their motivation and visuomotor skills were assessed by placing the platform 1 cm above the water level during a single session of 4 tests of 60 s maximum each (interval between tests: 30 min). A test began by placing the animal, facing the wall, at one of the four starting points (E, N, W, S). During the 4 consecutive days, the spatial learning abilities were tested by placing the platform in the NW quadrant. The animals had to use distal visual cues around the basin to find the location of the platform. Each day, 4 tests of 60 s maximum were performed. On day 4, 2 h before the end of the fourth trial, a probe test was performed by removing the platform and recording mice swimming into the basin for a unique trial of 60 s. Spatial memory capacities were evaluated 3 days after the end of the training phase, during the retrieval test. Mice were placed in the basin according to the protocol described above. Behavioral flexibility was evaluated for 4 consecutive days by moving the platform initially from the NW quadrant to the SE quadrant. The procedure was the same as described above. Platform location latency, distance traveled and swimming speed were measured.

The animal search strategies during spatial learning to find the platform in the MWM were analyzed, as previously validated [34,35]. Briefly, from the Anymaze software (Stoelting Co., Wood Dale, IL, USA), the image files plotting the swimming trajectories employed by each mouse in the 4 trials of the 4 days of spatial learning (D9–D12) or in trials of the long-term memory testing day (D15) were extracted into the Microsoft Windows Bitmap Image format (BMP, 140 × 120 pixels, 32 bits), then converted to a reduced-size grayscale format (48 × 48 pixels) using a free image processing tool of Python interpreter (Pillow, Alex Clark and Contributor). The most representative images were manually labeled according to six categories of strategies: thigmotaxis, scanning, circling, focal search, rotating, or direct swim. A schematic representation of each strategy is shown in Figure 2D. We used a validated strategy recognition model [33] to analyze our unlabeled images and automatically identify the main strategy per trial and per mouse. Then, a score was extracted from the different strategies: Thigmotaxis (0) < Scanning (1) < Circling (2) < Focal search (3) < Rotating (4) < Direct swim (5). The cognitive score for each day was determined by averaging the scores obtained across the 4 trials of the day as a sign of cognitive performance [35,36,37]

### 2.9. Spontaneous Activity, Nycthemeral Cycle and Anhedonia

In animals receiving treatment regiments and protocols of Sub-study 1, anhedonia (loss of the ability to feel pleasure) was evaluated with the sucrose preference test [38]. Mice were isolated in individual cages the day before the test (dimensions: 17 × 28 cm on the floor) with enrichment (lignocel^®^, JRS, Rosenberg, Germany). They had access to two bottles, one containing water and the other one containing a 2% sucrose water solution (Sigma Aldrich, S5016, Saint-Quentin-Fallavier, France). Consumption in each of these bottles as well as food intake and weight gain or loss were measured on a daily basis for 4 days. The sucrose preference index was established according to the following calculation: volume of sucrose/(volume of sucrose solution + volume of water).

Spontaneous activity and the circadian cycle were then studied in the same animals individually recorded (11 × 21 × 17 cm) into a dedicated actimetry device. During 4 days, locomotor and vertical activities were recorded by an actimeter (Imétronic^®^, Pessac, France) using infrared sensors placed around the cages.

### 2.10. Preparations of Tissue Samples

On the day following the completion of the treatment and behavioral period (Sub-study 1) or immediately after the last treatment administration (Sub-studies 2, 3 and 4), mice were anesthetized using isoflurane and either sacrificed by decapitation, for sampling of the brain (Study 1) frozen in isopentane cooled to −30 °C and then stored at −80 °C, or subjected to intracardiac puncture of blood in heparin-coated tubes (Greiner Bio One, 454084) for plasma (Sub-study 4) or in Serum Clot Activator Tubes (Greiner Bio One, 454204) for serum (Sub-study 3) collection. Plasma was recovered after centrifugation (1500 rcf, 15 min, 4 °C) and stored at −80 °C. Serum was recovered after centrifugation (2000× *g*, 10 min, 4 °C) and stored at −80 °C. In Sub-studies 2, 3 and 4, mice were decapitated, brains and small intestine were sampled, and the intestinal content was removed by rinsing in sterile PBS and immediately frozen.

To collect the cecal content (Sub-study 3), the digestive tract was isolated during necropsy, the cecum was clamped upstream and 1 cm downstream of the cecum, and the digestive tract was cut on both sides to collect the cecum in a sterile dish. The cecal content was obtained by pressing the cecum and transferred into a sterile 2 mL Eppendorf tube kept on ice. The weight of the cecal content was measured and rapidly stored at −80 °C for gut microbiota analyses.

### 2.11. Gut Microbiota Analyses

DNA content of cecum was extracted using the QIAamp DNA Stool Mini Kit (QIAGEN), including a bead-beating step (0.1 mm zirconia silica beads, BioSpec products, Bartlesville, OK, USA) [39]. To quantify bacterial taxa, quantitative real-time polymerase chain reactions (qPCR) were performed on the extracted DNA using Itaq Universal SYBR Green Supermix (BioRad, Marnes–La–Coquette, France) and 16S rRNA specific primers (primer sequences are detailed in Appendix A; see corresponding references [40,41,42,43,44,45]). qPCR signals were detected on a Mastercycler ep Realplex system (Eppendorf, Hamburg, Germany). All samples were run in duplicate in 96-well reaction plates. The final concentrations were as follows: DNA 0.25–2.5 ng/µL (depending on the targeted taxon), primers 0.5 µM and Itaq Universal SYBR Green Supermix 1X. The thermocycling conditions were as follows: initiation step at 95 °C 5 min; cycling stage at 95 °C 5 s, 30 s at annealing temperature (Appendix A), 40 cycles; melt curve stage at 95 °C 15 s, 65 °C 15 s, increment of 1 °C every 10 s until reaching 95 °C. The purity of the amplified products was verified by analyzing the melt curve performed at the end of amplification. At least 85% of the duplicates showed a variation lower than 0.5 Cq units. Serial dilutions of DNA from the cecal contents were included on each plate to generate a relative curve and to integrate the primer efficiency in the calculations. Non-template controls were included on each plate. A qPCR was considered valid if the Cq of the non-template control was at least 3 units higher than the Cq of the templates. For detection of total *Eubacteria*, the Cq of each sample were compared with a standard curve produced by diluting the genomic DNA extracted from a pure culture of *E. coli*, for which the cell counts were determined by plating and quantifying the colony-forming units.

### 2.12. Ginsenosides HPLC Analysis

Analyses of serum were performed on a Waters HPLC system (binary pump model 1525, DAD model 996, autosampler CTC PAL AOC-5000) coupled with a mass spectrometer ZMD micromass from Waters (Milford, MA, USA). Acetonitrile, trifluoroacetic acid and methanol were purchased from VWR (Radnor, PA, USA). Ginsenoside Rb1 was purchased from Sigma (Saint-Louis, MO, USA), and Ginsenosides Rg1, Re, Rc, Rd, K, Protopanaxadiol, Protopanaxatriol standards were purchased from Extrasynthese (Genay, France).

The extraction of ginsenosides from mouse serum was adapted from the publication of Kim et al., 2013 [46]: 400 µL of refrigerated acetonitrile at −20 °C was added to 100 µL of mouse serum. The solution was vortexed and centrifuged at 7500× *g*, 5 min, then lyophilized with a Thermo Scientific PowerDry LL 1500 (Waltham, MA, USA). The sample was resolubilized in 50 µL methanol. The ginsenosides were separated from the samples on an HPLC column Agilent Poroshell EC-C18 (100 × 3 mm Ø2.7 µm). A gradient with solvent A (water HPLC grade with 0.05% trifluoroacetic acid) and solvent B (acetonitrile HPLC grade with 0.05% trifluoroacetic acid) was used at a flow rate of 0.3 mL/min. The initial step was 20% solvent B. The solvent gradient was 0–20 min 53% B, 30 min 100% B, 35 min 100% B, 36 min 20% B, 45 min 20% B. The volume injection of the sample was 20 µL and of the standard was 10 µL.

The electrospray ionization source of the mass spectrometer was operated in the negative (ESI−) and positive (ESI+) ion mode. Detection was performed in ESI− at *m*/*z* 914 (Rg1/Rf/F11; [M + TFA-H]−), *m*/*z* 1060 (Re/Rd; [M + TFA]−), *m*/*z* 898 (Rg2/Rg3/F2; [M + TFA]−), *m*/*z* 1222 (Rb1; [M + TFA]−), *m*/*z* 1192 (Rc/Rb2/Rb3; [M + TFA]−), *m*/*z* 1070 (Ro; [M + TFA]−), m/z 880 (Rf4/Rk1/Rg5/Rg6/Rg4; [M + TFA]−). The detection in ESI+ target *m*/*z* 425 (protopanaxadiol and ginsenoside K) and *m*/*z* 423 (protopanaxatriol). The identification of ginsenosides was compared to the publication of Park et al., 2013 [47].

For the positive ESI analysis, the parameters were as follows: Capillary 3.50 kV; Cone 40 V; Extractor 5 V; RF Lens 0.30 V; Source Block Temperature 120 °C; Desolvation Temperature 250 °C; Desolvation gas 6.7 L/min. For the negative ESI analysis: Capillary 3 kV; Cone 50 V; Extractor 3 V; RF Lens 0.50 V; Source Block Temperature 120 °C; Desolvation Temperature 250 °C; Desolvation gas 6.7 L/min.

### 2.13. Cytokines Elisa Quantification

The plasma concentrations of cytokines were analyzed by the enzyme-linked immunosorbent assay (ELISA) according to the manufacturer’s recommendations (Q-Plex Mouse Cytokine-Inflammation, Quansys Biosciences, Logan, UT, USA). The cytokines measured were: TNF-α, IL-1β, IL-1α, IL-6, IL-12, IL-2, IL-4, IL-10, IL-17, IL-3, Granulocyte Macrophage Colony-Stimulating Factor (GM-CSF), Monocyte Chemotactic Protein-1 (MCP-1), Macrophage Inflammatory Protein-1 (MIP-1α) and Regulated on Activation Normal T cell Expressed and Secreted (RANTES). The chemiluminescence from the 96-well plates was imaged with ChemiDoc XRS+ (Bio-Rad, Marnes–La–Coquette, France) and an 8-point calibration curve (7 points plus 1 blank); the controls and samples were assayed in duplicate. Then, the images of the Q-plex plates were analyzed in Q-ViewTM software (Quansys Biosciences) to obtain the standard curves and sample values in pg/mL for the samples. The concentrations of IL-10 and MIP-1α were not determined because the pixel level of the zero point for the standard curve was higher than the pixel intensity of the samples in all the treatment groups. All data are expressed in pg/mL. Undetectable values are expressed as half of the minimal quantity detected by the kit within this experimental sequence. Statistical quantification among the 6 groups was assessed using the Kruskal–Wallis test followed by Dunn’s multiple comparison test. Data are expressed as mean ± SEM.

### 2.14. Histological Analysis of Intestinal Sections

Sections of 20 µm thickness of small intestine were cut onto gelatin-chrome alum coated slides and stained with hematoxylin and eosin (H&E). Briefly, the sections were washed for 15 s with 95% ethanol (EtOH) and then fixed during 10 s with 10% formalin. After washing for 10 s with demineralized water, the slices were immersed for 10 s in a hematoxylin solution (Sigma-Aldrich, HHS32). Then, the sections were washed twice for 10 s with demineralized water, followed by a 10 s wash with 95% EtOH, before being incubated for 10 s in an eosin solution (Sigma-Aldrich, HT110232). Finally, the sections were washed by two consecutive immersions in EtOH 95%, EtOH 100% and xylene solution (Fisher Chemical X/0250/17), during 2 × 10 s each. Histological changes, such as the intestinal surface area and villi length, were imaged and monitored under a ZEISS Axioscope 7 microscope (PRMACEN platform, University of Rouen Normandie). The size of the villi length and the surface area were analyzed using computer-based image analysis software (version 1.8.0, NIH, MD, USA).

### 2.15. Immunohistochemical Analysis of Brain and Intestinal Tissues

Small intestines and brains were cut into 20 μm thick serial coronal sections with a cryostat. Intestinal sections were mounted on slides coated with gelatin-chrome alum (VWR International, Leuven, Belgium) and stored at −20 °C until processing. Brain sections were cut from the anterior part of the dorsal hippocampus (anteroposterior, 1.08 mm from the Bregma −1.46 mm). Every 12th section, each separated by 240 mm, was mounted on slides coated with gelatin-chrome alum and stored at −20 °C until processing. Eight hippocampal sections from three to four animals per group were stained simultaneously for immunohistochemical observations and quantifications. Slices were post-fixed with formaldehyde (4%, Sigma-Aldrich) for 15 min at 4 °C, rinsed in PBS (0.1 M, pH 7.4, Sigma-Aldrich), permeabilized with X-100 triton (Fisher Scientific, Illkirch, France) 0.05%, and non-specific binding sites were blocked with a permeabilization/blocking solution containing PBS (Sigma-Aldrich), 0.5% BSA (Roche, MannHeim, Germany), 0.1% X-100 triton and 10% NDS (Normal Donkey Serum, Sigma-Aldrich) for 60 min at room temperature (RT). For BrdU immunostaining, brain slices were first incubated in a 2N-HCl for 45 min at 45 °C and then rinsed in a boric acid (Sigma-Aldrich) solution 0.1 M at pH = 8.5 for 10 min. Brain slices were incubated overnight (12 h) at 4 °C, with the primary antibodies of interest diluted in the permeabilizing/blocking buffer (PBS, 0.5% BSA, 0.1% X-100 triton and 1% NDS).

Brain sections were incubated overnight at 4 °C with the following primary antibodies: sheep anti-BrdU (1:100, Abcam, Paris, France, ab1893), rabbit anti-NeuN (1:1000, Abcam, ab104225), goat anti-glial fibrillary acidic protein (GFAP) (1:1000, Abcam, ab53554) or rat anti-IL-6 receptor gp130 (1:200, Invitrogen, Boulogne-Billancourt, France, MA5-23922) in the 10% blocking solution. Then, sections were rinsed (3 × 10 min) and incubated for 2 h at room temperature with fluorescent-dye-conjugated secondary antibodies (1:1000): Alexa 647-conjugated donkey anti-sheep (Abcam, ab150179), Alexa 488-conjugated donkey anti-rabbit (Invitrogen, A21206), Alexa 568-conjugated donkey anti-goat (Invitrogen, A11057) or Alexa 647-conjugated donkey anti-rat (Abcam, ab150155) in PBS. Intestine sections were incubated overnight at 4 °C with the primary antibody rat anti-monocyte/macrophage (MOMA-2) (1:100, Abcam, ab33451) and incubated for 2 h at room temperature with fluorescent-dye-conjugated secondary antibodies Alexa 647-conjugated donkey anti-rat (Abcam, ab150155) in PBS. After rinsing, the slices were co-stained with 4′,6-diamidino-2-phenylindole (DAPI), a nuclear and chromosome counterstain, for 10 min. The sections were then embedded with Mowiol. Images were captured using a Leica SP8 Confocal Microscope (Leica Microsystems, Nanterre, France, Primacen Platform).

For immunofluorescence quantification, the number of BrdU-, BrdU/NeuN- or gp130-positive cells was counted in the dorsal (bregma from −1.34 to −2.30 mm) and ventral (bregma from −2.92 to −3.28 mm) dentate gyrus of the Hp. These brain regions were identified by referring to the Paxinos and Franklin’s atlas. For each analyzed mouse, positive cell counts and measured surface areas of the dentate gyrus were performed on 4 sections of the dorsal Hp and 4 sections of the ventral Hp. The density of cells in the dorsal or ventral dentate gyrus per mouse was evaluated by the following formula: Density (cells/mm^2^) = ∑ nb cells in 4 section/∑ area in 4 section.

### 2.16. Statistical Analysis

All statistical analyses were performed using R (version 4.0.0, http://r-project.org, accessed on 20 July 2022). A probability value of *p* < α(0.05) was considered statistically significant (* *p* < 0.05; ** *p* < 0.01; *** *p* < 0.001). When the data followed a normal distribution (as assessed by the Shapiro test), a one-way ANOVA parametric test was performed. In the case of repeated measures, a two-way ANOVA was performed to compare the main effects (treatment comparisons) with the day as the repeated measure and the dependent variable (latency, distance, swimming speed, activity or weight gain). A Bonferroni test was used for *post hoc* analyses. In the absence of normal data distribution, a non-parametric Kruskal–Wallis test was performed to compare the different groups. In this case, Dunn’s test was used for the post hoc analyses. Kendall’s rank correlation was used to analyze the strength of association between the variables.

For a cognitive score of the MWM test, the classification of strategies employed by mice in the MWMT was evaluated by the semi-automated method described by Higaki et al. [35]. SciPy module, the open-source scientific tool provider for Python, was used for this statistical analysis.

For microbiota statistical analyses, the outliers were identified using the ROUT method with a false discovery rate Q = 1% [48]. Due to the absence of normal data distribution, comparisons of bacterial taxa levels were performed using the non-parametric Kruskal–Wallis test with Dunn’s multiple comparison test (uncorrected for the analysis of *Lactobacilli* and its subgroups). A probability value of *p* < 0.05 was considered statistically significant.

## 3. Results

### 3.1. Deleterious Impact of 5-FU and Qiseng^®^ on Spontaneous Activity, Nycthemeral Activity and Exploration

Here, we evaluated the behavioral consequences of 5-FU chemotherapy, associated with the *per os* administration of the placebo, vitamin C or Qiseng^®^, for the general spontaneous and circadian activity, exploration, emotional reactivity, spatial learning and memory in mice by means of validated behavioral tests (Figure 1A).

The weights of the animals from the first day of the experimental period are displayed in Figure 1B. Mice receiving NaCl at D0, D7 and D14 (red arrows) and placebo (solid blue line), vitamin C (solid green line) or Qiseng^®^ (solid red line) showed an increasing weight gain over the treatment period. As revealed by the repeated-measures bi-directional ANOVA (iteration: F80,2096 = 11.55), there was no difference between the control (non-chemotherapy) groups. The administration of 5-FU on D0, D7 and D14 triggered an expected [49] weight loss of 3% to 5% on the day following the injection in placebo (dotted blue line), vitamin C (dotted green line) or Qiseng^®^ (dotted red line) groups. 5-FU mice partially regained weight at D7 and D14, the weight of 5-FU-treated mice being lower than the control (NaCl-treated) mice from the second 5-FU injection until the end of the treatment period (placebo: *p* = 0.01, vitamin C: *p* = 0.009, Qiseng^®^: *p* < 0.001). Neither vitamin C nor Qiseng^®^ compensated this 5-FU-induced weight loss (Figure 1B). It can be noticed that the dosage of 60 mg/kg once weekly for 3 weeks, combined with daily gavage, caused death in some mice in the various groups of 5-FU-treated mice (Figure 1C). A lower percentage of mortality (4%) was observed in 5-FU mice receiving Qiseng^®^ compared with 5-FU mice receiving placebo (11.5%) or vitamin C (16%).

In order to evaluate the locomotor activity of the animals under chemotherapy in a closed and unknown environment, we used the OFT two days after the last administration of placebo, vitamin C or Qiseng^®^. The OFT showed no significant differences for the control mice subjected to the three different gavage treatments with regard to the distance traveled, immobility time or vertical activity, suggesting that neither vitamin C nor Qiseng^®^ altered spontaneous activity (Figure 1D). In contrast, the distance traveled by 5-FU/placebo or 5-FU/vitamin C mice was lower compared with NaCl mice. 5-FU-treated mice receiving Qiseng^®^ exhibited similar locomotor activity, total time of immobility and vertical activity (exploration) to control mice (Appendix A), suggesting a preventive/compensatory role of Qiseng^®^ in the effects of 5-FU on animal welfare and activity (Figure 1D). When analyzing the distance traveled in the center or in the periphery of the device, we observed that mice treated with 5-FU/placebo and 5-FU/vitamin C traveled less distance in the center than the control mice (*p* = 0.05, Figure 1D) (Appendix A). Interestingly, 5-FU/Qiseng^®^ mice showed statistically similar exploration in the central and peripheral area as the control mice (Appendix A). These data show that the Qiseng^®^ solution administered during and after chemotherapy compensates the deleterious effects of 5-FU on locomotor activity and exploration.

Because the post-chemotherapy fatigue symptom may be explained by a disturbance in the sleep/wake rhythm [50,51], the impact of the different treatments was assessed during the nycthemeral cycle, the activity being evaluated for 4 consecutive days (actimetry test, Figure 1E). The active behavior of the C57Bl/6 mouse strain can be divided into two peaks of activity: between 12 and 20 h, and between 20 and 00 h [52]. These mice showed an absence of activity during the light phase (00–12 h) and adopted an active behavior during the dark phase (12–00 h). All animals, including the 5-FU groups, retained these periods of activity or inactivity according to the light/dark phase without an apparent shift between the treated groups. They showed a similar period of much reduced activity during the light phase (Figure 1E) compared to the dark one. The 5-FU/placebo (*p* < 0.001) or 5-FU/vitamin C (*p* = 0.004) mice exhibited a significant reduction in activity during the dark phase, at both activity peaks (12–20 h and 20–00 h) in 5-FU/placebo and during the first activity peak (12–20 h) in 5-FU/vitamin C mice. The 5-FU/Qiseng^®^ mice did not show this decreased activity (peak 1 and 2) during the dark phase (Figure 1E). Thus, chemotherapy inhibited the activity during the awake phase without affecting the night phase or the sleep/wake cycle phase. The Qiseng^®^ solution fully prevented the chemotherapy-induced decrease in activity for several days.

In order to test the role of chemotherapy and Qiseng^®^ in mouse activity in the context of an aversive/anxious situation, we tested the mice in the emergence test using a LDB. This test is based on the rodents’ innate aversion to brightly lit areas and on their spontaneous exploratory behavior in response to mild stressors, e.g., a new environment and light. Switching from one compartment to another gives an indication of the exploratory activity associated with addiction/motivation over time and the time spent in each compartment as a reflection of aversion [31]. The 5-FU/placebo or 5-FU/vitamin C animals showed a significant decrease in both the number of entries into the lighted area (*p* < 0.001 and *p* = 0.01) and the time spent (*p* < 0.001 and *p* < 0.02) in the lighted compartment compared to the controls. Mice treated with 5-FU/Qiseng^®^ did not show significant differences in these parameters compared with the controls. The groups NaCl/placebo/ or NaCl/vitamin C, or NaCl/Qiseng^®^ mice did not show significant differences in these parameters (Figure 1F). These results indicate that chemotherapy led to diminished exploratory activity in aversive situations but that the mice receiving both chemotherapy and Qiseng^®^ expressed a better adaptability and response to mild stressors (new environment and light), similar to the control mice, suggesting prevention and/or compensation.

### 3.2. No Effect of Treatments on the Emotional Reactivity of Chemotherapy Mice

It has been previously observed that chemotherapy promotes anxiety- and anhedonia-like behavior in rats [53]. Here, to evaluate the anxiety state, we used the EPM test 3 days after the last administration of placebo, vitamin C or Qiseng^®^ and 11 days after the last injection of NaCl or 5-FU (Figure 1A). In this test, the total distance traveled, total immobile time, number of stretched attend postures (SAP) and head dips did not differ significantly between the different groups of animals (Appendix A). Similarly, the time spent and distance traveled in each arm was similar between the groups (Appendix A). The action of the administered compounds on the resignation behavior, a condition associated with depression, was then assessed by the TST and FST (Figure 1A). In these two tests, the latency of first immobility and total immobility time were similar between the different groups, reflecting the absence of depressive-type behavior in the animals (Appendix A). In addition, a sucrose preference test was performed to assess whether the mice suffered from anhedonia—a lack of willingness to seek pleasure (reward circuit). There was no difference between the groups, confirming the absence of anhedonia (Appendix A). As previously established, the administration of 5-FU in mice does not induce anxiety or depressive-like behavior [49]. We therefore showed that neither the 5-FU nor vitamin C and Qiseng^®^ treatments resulted in the development of anxiety or depressive-like behavior (resignation or anhedonia) in all the treated groups. 

### 3.3. Impact of 5-FU Chemotherapy on Spatial Learning, Memory and Flexibility

The cytotoxic drugs, notably 5-FU, currently used in the treatment of cancers have been shown to have an impact on spatial memory and/or behavioral flexibility [54]. In order to examine the deleterious effects of 5-FU on the cognitive processes and the potential protective effects of Qiseng^®^ or vitamin C treatments on the cognitive processes, the MWMT was used to assess spatial learning, memory and learning flexibility (Figure 2A). On day 8, the visible platform was placed in the center of the device (Figure 2A), and we observed that the platform location latency, distance traveled and swimming speed were similar across the different groups (Appendix A). These results indicate that the visual and movement ability in the device was identical between the different groups of animals. During the learning phase, the 4-day training resulted in a significant decrease in platform location latency in all groups (Figure 2B). Analysis using a bi-directional repeated-measures ANOVA (day effects: F3.786 = 118.8) showed that all 5-FU mice exhibited significantly higher platform location latency than the control mice (placebo: *p* = 0.01, vitamin C: *p* = 0.02, Qiseng^®^: *p* < 0.001), while the swimming speed did not differ between the animals (Appendix A). Thus, chemotherapy affects the learning ability, leading to difficulties to associate the visual cues with the location of the platform. This learning difficulty of 5-FU mice did not persist over time, as they behaved similarly to the control groups on the last two days (D11 and D12) (Figure 2B). Neither vitamin C nor Qiseng^®^ compensated the deleterious effects of chemotherapy on spatial learning abilities. Taken together, our results show that chemotherapy has an impact on spatial learning 20 days after the last injection. This effect was compensated neither by vitamin C nor by Qiseng^®^.

In order to evaluate the animals’ short-term spatial memorization, a test probe session was conducted at the end of the last learning session on day 12 (Figure 2A). Statistical analyses revealed no difference in the percentage of time spent or distance traveled in the NW quadrant by the different groups of animals (Appendix A). Therefore, since all mice in the different groups spent the same amount of time in the quadrant where the platform was initially located, it appeared that all animals correctly memorized the location of the platform at the end of the training session. Thus, chemotherapy and the treatments did not impact short-term memorization of the location of the platform in the device.

In an effort to assess the long-term memorization of the animals, a recall session was conducted on day 15, 2 days after the end of the last learning session on day 12 (Figure 2A). Statistical analysis showed that all 5-FU mice exhibited a higher platform locating latency and a greater swimming distance traveled than the control mice (all *p* < 0.001) (Figure 2C). The swimming speed remained the same between the different groups of mice (Appendix A), suggesting identical locomotor abilities. We therefore observed that 5-FU mice searched the platform for a longer period of time, with an increase in the distance traveled in the pool (Appendix A), suggesting altered long-term spatial memorization. Neither vitamin C nor Qiseng^®^ had a beneficial effect on the deleterious effects of 5-FU on long-term spatial memorization (Figure 2C). In addition to these results, we used a Python neural network to quantify the six different swimming strategies used by mice (Figure 2D) during the learning and recall phase sessions, as described in Higaki et al. [35]. The cognitive score obtained for each group of mice during the spatial learning phase was the same (Figure 2E), indicating that all mice adopted the same swimming strategy during this phase. Interestingly, 5-FU induced a decreased cognitive score in the long-term memory assessment on D15 (Figure 2F), highlighting a less effective strategy used by mice under chemotherapy. Statistical analysis using the Pearson residue test indicates that during this session, all NaCl control mice used only the most efficient strategy of “direct swim” (Figure 2F). NaCl/Qiseng^®^ mice exhibited a higher frequency of this swimming strategy than NaCl/placebo and NaCl/vitamin C mice. In mice undergoing chemotherapy, the results indicate a statistical absence of this direct swim strategy, confirming the observed impairment of long-term spatial memory. Indeed, 5-FU/placebo and vitamin C mice employed a less effective strategy, mainly “Focal search” combined with “Scanning” for 5-FU/placebo and “Circling” for 5-FU/vitamin C. Although they also showed long-term memory impairment, the 5-FU/Qiseng^®^ mice employed a platform location strategy known as “Rotating”, which was more effective than those used by the other two groups of 5-FU mice (Figure 2F), thus suggesting a beneficial effect of Qiseng^®^ in the neurobiological mechanisms supporting learning and memory. 

Flexibility on D16 was evaluated as a function of the platform location latency, distance traveled and swimming speed. All 5-FU mice showed a statistically significant increase in platform location latency (placebo: *p* < 0.001, vitamin C: *p* = 0.002, Qiseng^®^: *p* < 0.001) (Figure 2G), as well as in the swimming distance traveled compared with NaCl mice (Appendix A). The swimming speed did not differ between the groups. These results highlighted the difficulty of 5-FU mice in unlearning information. The performance of mice in relocating the platform was then measured for 3 consecutive days (D17-19) without changing the platform, constituting a new learning session. In accordance with the results obtained in the first learning session (D9–D12), the analysis using a bi-directional ANOVA (day effects: F3.786 = 112.8) with repeated measurements showed that 5-FU mice had a higher platform relocation latency than NaCl-treated mice (all *p* < 0.001) (Figure 2G). These results are indicative of a difficulty for all mice undergoing chemotherapy in relearning the platform location. Together, we established here that the 5-FU mice presented a greater difficulty in spatial learning and long-term memory. An alteration in learning flexibility was also found in these mice. Neither vitamin C nor Qiseng^®^ significantly prevented these deleterious effects.

### 3.4. Impact of 5-FU Chemotherapy on Hippocampal Neural Stem Cells Proliferation and on Neurogenesis

The radar plots in Figure 3A showed that the administration of 5-FU decreased the spontaneous locomotor activity, exploration and altered spatial learning, memory and flexibility. The co-administration of Qiseng^®^ prevented the deleterious effects of chemotherapy on activity but not on cognition. 5-FU has been shown to be associated with the inhibition of proliferating neuronal precursors and/or the integrity of mature neurons [54]. The administration of 5-FU and leucovorin to rats resulted in decreased survival of the dividing cells in the subgranular area of the dentate gyrus immediately after treatment, worsened over the next six weeks [55]. To monitor the effect of 5-FU on neurogenesis, we studied the dorsal and ventral Hp brain areas of mice previously used in behavioral tests. Indeed, dHp and vHp are found involved in the control of spatial memorization and mood/stress behavior, respectively [56]. By counting both BrdU and NeuN positive cells, in all chemotherapy groups, we found a significant reduction in NeuN/BrdU+ cell density in the dorsal gyrus compared with their controls (*p* < 0.05). The 5-FU/placebo (*p* = 0.01) and 5-FU/vitamin C (*p* = 0.03) mice showed, in addition, a decrease in the density of NeuN/BrdU+ cells in the vHp compared with their controls. Interestingly, 5-FU/Qiseng^®^ showed similar NeuN/BrdU+ cell density as NaCl/Qiseng^®^ mice in the vHp (Figure 3B). In a different experiment, placebo, vitamin C and Qiseng^®^ were administered in the two last weeks of the 5-FU treatment period. Then; mouse brain, were collected (Figure 3C). From the brain slices, the analysis of neural stem cell proliferation in the Hp was evaluated by quantifying the number of positive BrdU cells in the gyrus. The 5-FU/placebo and 5-FU/vitamin C mice showed a reduction in the density of BrdU positive cells in the dorsal (*p* = 0.01 and *p* = 0.06) and ventral (both *p* = 0.01) Hp compared with their controls. The 5-FU/Qiseng^®^ mice showed a similar proliferation to the NaCl/Qiseng^®^ mice in these two regions (Figure 3C). These data reveal that Qiseng^®^ prevents a chemotherapy-induced alteration of the proliferation of neural precursor cells in the Hp and of neurogenesis in the vHp specifically.

### 3.5. Detection of Serum Ginsenosides in Mice Receiving Qiseng^®^

Ginseng in general and *Panax quinquefolius* in particular are composed of Ginsenosides distributed in protopanaxadiol-type (Rb1, Rc, Rd, Rg3, F2, Rb2, etc.), protopanaxatriol-type (Re, Rg1, F1, Rh1, etc.) and oleanane-type ginsenosides (Ro) [57]. The pseudoginsenoside F11 is, in addition, specifically found in the American ginseng [20]. The circulating levels of ginsenosides in the control or 5-FU groups were evaluated on the day following the completion of treatments (Figure 4A). HPLC analysis of the solution of Qiseng^®^ used for administration in mice showed the presence of ginsenosides Rb1, Re, F11/Rf, Rg2, Rb1, Rc, Ro, Rb2, Rb3, Rg5, Rk1 and Rf4 (Figure 4B). After two weeks of treatments (Figure 4A), ginsenosides—mainly Rb1, Rd and Rc—were found in the serum of NaCl/ and 5-FU/Qiseng^®^ groups (Figure 4B,C). No ginsenosides were detected in the serum of placebo- or vitamin-C-fed mice (data not shown). Interestingly, the relative percentage of ginsenoside Rc was significantly increased in the serum of mice treated with 5-FU (+13.38%; *p* = 0.03). It was previously established that Rb1 and Rc can be metabolized by gastric acid and/or intestinal bacteria into smaller compounds, such as Rd and compound K, and further into Protopanaxadiol (Figure 4D) [58]. Here, we failed to detect compound K in all serum samples, but the proportion of Protopanaxadiol was detected in serum and was found similar between the NaCl/Qiseng^®^ (0.03 ± 0.011 mg/L) and 5-FU/Qiseng^®^ groups (0.028 ± 0.005 mg/L) (Figure 4E).

### 3.6. Impact of 5-FU Chemotherapy and Qiseng^®^ on the Intestinal Microbiota

The modification of the gut microbiota composition may have important consequences for intestinal inflammation [59]. It may also impact behavior by altering the gut–brain axis [60]. Interestingly, 5-FU has previously been shown to increase Verrucomicrobia and decrease Firmicutes in BALB/c mouse cecum, resulting in severe inflammation of the intestinal mucosa associated with weight loss and infiltration of inflammatory cells [61]. We evaluated the impact of 5-FU on the abundance of several bacteria taxa in the gut microbiota of mice on the day following the completion of treatment (Figure 5A). The bacterial domains, phyla, classes, genera and species are shown in Figure 5B. We observed that 5-FU alters gut microbiota composition and specifically modulates Lactobacilli (including *L. murinus/animalis*, *L. acidophilus* and *L. johnsonii/gasseri*). The other monitored taxa levels remained unaffected by 5-FU (Figure 5C–G). Interestingly, the 5-FU/Qiseng^®^ mice did not show any alteration in Lactobacillus abundance, suggesting that Qiseng^®^ prevents, at least in part, the dysbiosis induced by 5-FU. The 5-FU/Qiseng^®^ mice exhibited a significant decrease in Archae and Verrucomicrobia compared with the 5-FU/placebo and 5-FU/vitamin C groups (Figure 5C,D), and all mice under Qiseng^®^ (NaCl and 5-FU) showed a significant decrease in Eubacteria (*p* < 0.01) and in Beta-, Gamma- and Delta-proteobacteria (*p* < 0.001) compared with the placebo and/or vitamin C groups (Figure 5C,G). Together, these results indicate that Qiseng^®^ may partially compensate the gut microbiota dysbiosis triggered by 5-FU treatment. To determine any similarity or disparity among our different mouse groups, a PCA representation of microbiota composition was performed (Figure 5H, on the left). No opposition or tight convergence was observed between the vectors, reflecting a lack of correlation between the groups of bacteria. However, the 5-FU/placebo and 5-FU/vitamin C samples were mostly arranged along the Lactobacillus vector. Graphically, the NaCl/Qiseng^®^ and 5-FU/Qiseng^®^ data were grouped together and opposed to all vectors. To validate these distributions, cluster determination was performed by calculating the relative cluster stability index (RCSI) using the elbow method in complement, both establishing the optimal number of clusters as three (Figure 5H, on the right). Cluster 1 consisted mainly of all NaCl/Qiseng^®^ (*n* = 10) and 5-FU/Qiseng^®^ (*n* = 8) samples. Cluster 2 included the majority of the 5-FU/placebo (*n* = 9), 5-FU/vitamin C (*n* = 9) and 5-FU/Qiseng^®^ (*n* = 3) samples. Finally, the very large Cluster 3 contained the NaCl/placebo (*n* = 5) and NaCl/vitamin C (*n* = 8) samples. These results indicate that the Qiseng^®^ group (Cluster 1) was opposed to the increase in the different bacterial taxa and correlated with the decrease in some of them, notably the Proteobacteria classes. On the other hand, the 5-FU/placebo and 5-FU/vitamin C groups could be statistically combined into Cluster 2, part of which was arranged along the Lactobacillus vector, showing the impact of 5-FU on the increase in this bacterial genus (Figure 5H, on the right).

### 3.7. Impact of Chemotherapy and Qiseng^®^ on Intestinal Integrity

Chemotherapy is a known inducer of cell reactive oxygen species (ROS), whose production can directly induce intestinal tissue damage and trigger the activation of the NF-κB factor responsible for the expression of pro-inflammatory cytokines, such as TNF-α, IL-1β and IL-6 [16,17]. We first evaluated the effect of 5-FU on the morphology of the intestinal mucosa. The administration of 5-FU chemotherapy (60 mg/kg/week for 3 weeks) (Figure 6A) resulted in a significant decrease in the intestinal surface area (*p* = 0.02), not compensated by the co-administration of vitamin C (*p* = 0.03) (Figure 6B). However, this intestinal lesion was limited by the administration of Qiseng^®^ (*p* = 0.06). Additionally, 5-FU caused a marked decrease in intestinal villi height, which was compensated neither by vitamin C nor Qiseng^®^ (all *p* = 0.008) (Figure 6B). The quantification of intestinal inflammation by using a macrophage marker (MOMA-2) showed a stronger intestinal invasion by macrophages in all groups receiving chemotherapy compared with their respective NaCl controls (Figure 6C). The level of macrophages was lower in the 5-FU/vitamin C mice compared with the 5-FU/placebo mice (*p* < 0.001) and even more reduced in the 5-FU/Qiseng^®^ mice (*p* < 0.001). Thus, vitamin C and, more efficiently, Qiseng^®^, compensated immune cell infiltration in the intestinal mucosa in response to the 5-FU treatment. The 3D scatter plots of the relationship between the intestinal surface, villi height and macrophage infiltration showed a strong correlation between the diminished intestinal surface, villi height and increased intestinal inflammation in mice undergoing chemotherapy, notably the 5-FU/placebo group (Figure 6C). Vitamin C and, more significantly, Qiseng^®^, reduced this negative impact of 5-FU. Thus, 5-FU induced an inflammatory intestinal response correlated with a decrease in intestinal surface and villi height. Vitamin C and Qiseng^®^ administration both decreased the inflammatory response, with Qiseng^®^ having a stronger impact on inflammation reduction and intestinal integrity than vitamin C alone.

### 3.8. Impact of Chemotherapy and Qiseng^®^ on Pro-Inflammatory, Pluripotent, Chemotactic and Leukocyte Growth Cytokines

Several studies have revealed correlations between fatigue and cytokine levels, mainly IL-6, in cancer patients [14] and have been positively correlated with cognitive and somatic fatigue [62], particularly in cancer patients [14]. In mice, it was shown that a single intravenous injection of IL-6 produced an increased immobility time [63] and a significant decrease in locomotor activity [15] in OFT. Here, we monitored the plasma levels of several different cytokines (Figure 7A). The 5-FU/placebo mice showed a significant increase in the pro-inflammatory cytokines IL12p70 (+66%; *p* = 0.03) and TNF-α (+124%; *p* = 0.009), pluripotent cytokines IL-2 (+217%; *p* = 0.04), IL-4 (+58%; *p* = 0.001), IL-6 (+266%; *p* < 0.001) and IL-17 (+244%; *p* < 0.001), and chemotactic cytokines MCP-1 (+532%; *p* < 0.001). We also observed a decrease in RANTES level (−37%; *p* = 0.05) in the 5-FU/placebo mice compared with the controls (Appendix A, Figure 7B). In contrast, the plasma levels of IL-1α, IL-1β, IL-3 and GMCSF cytokines were statistically similar to the controls. The 5-FU/vitamin C mice also showed an increase in IL-6 (+237%; *p* = 0.02) and MCP-1 (+273%; *p* < 0.001) levels and a decrease in RANTES plasma level (−48%; *p* = 0.02) (Appendix A, Figure 7B). Interestingly, the 5-FU/Qiseng^®^ mice showed no significant difference in the cytokine plasma profile compared with the non-chemotherapy-treated groups. Together, chemotherapy led to an increase in several inflammatory cytokines in plasma, partially prevented by vitamin C, while completely blunted by Qiseng^®^ (Appendix A, Figure 7B).

It was previously shown that intestinal inflammation and, in particular, pro-inflammatory cytokine secretion are correlated with changes in the composition of gut microbiota [59,64]. We made a correlation study between the different bacterial taxa quantified in the cecum and plasma cytokine levels (Figure 7C). A positive correlation was shown between the increase in the archaea domain and the elevation of IL-6 and MCP-1 levels. Similarly, the increase in these two cytokines correlated with the increase in Lactobacillus, which, in turn, correlated with the elevation of IL-4 and the plasma decrease in RANTES. Finally, a positive correlation between the abundance of L.acidophilus and the plasma elevation of IL-6 and MCP-1 (Figure 7C) was found. Thus, an increase in the bacterial taxa Archaea, Lactobacillus and L.acidophilus was mainly correlated with the increase in IL-6 and MCP-1 levels. According to these results, it was proposed that *L. acidophilus* upregulates a number of effector genes encoding cytokines, such as IL-6 and CCL2 (MCP-1), via NF-κB and p38 mitogen-activated protein kinase (MAPK) signaling pathways in intestinal epithelial cell lines and mouse models [65]. Here, we pointed out that Qiseng^®^, by acting on the 5-FU-induced microbiota dysbiosis, attenuated intestinal inflammation and markedly reduced systemic inflammation and plasma release of the key cytokines, such as IL-6.

### 3.9. Correlation Analysis between Neuronal Proliferation, Hippocampus and Systemic Inflammation

Systemic inflammation is associated with neuroinflammation via innervation and/or neurohumoral mechanisms. In order to assess the link between systemic inflammation and, more particularly, IL-6 and brain neuroinflammatory status, we quantified the expression of the IL-6 receptor gp130 (Figure 8A,B) within the Hp. Indeed, after ischemic injury, gp130 expression was shown enhanced in astrocytes, preferentially in the CA1 and dentate gyrus region [66]. In our 5-FU-treated mice, a significant increase in the density of the gp130 receptor was observed co-localizing with the NeuN+ neurons and GFAP+ astrocytes (white arrows) in the dHp (all *p* = 0.01) and vHp (all *p* = 0.01) compared with the control mice. The 5-FU/vitamin C mouse brains (910 ± 27.30 gp130+ cells/mm^2^) showed a lower density of gp130 labeling in Hp compared with in Hp of 5-FU/placebo (1074 ± 33.78 gp130+ cells/mm^2^; *p* = 0.01) brains. Interestingly, the gp130 expression in the vHp of the 5-FU/Qiseng^®^ mice (557.6 ± 40.26 gp130+ cells/mm^2^; *p* = 0.01) was lower than that of the 5-FU/placebo (199.5 ± 73.14 gp130+ cells/mm^2^; *p* = 0.01) and 5-FU/vitamin C (761.7 ± 46.43 gp130+ cells/mm^2^; *p* = 0.03) groups (Figure 8C). A correlation study of neural stem cell proliferation, Hp inflammation (gp130 expression) and plasma cytokine levels was performed (Figure 8C). In the placebo group, 5-FU evoked a decreased neuronal proliferation in both the dHp and the vHp, correlating with increased levels of IL-6. 5-FU also reduced cell proliferation in the vHp, correlating with enhanced MCP-1 plasma level. In contrast, no correlation was found between the plasma cytokines and proliferation in the Qiseng^®^ group (Figure 8C). Importantly, the gp130 receptor expression in the dHp and the vHp was positively correlated with plasma IL-6 in both the placebo and vitamin C groups treated with 5-FU. In the Qiseng^®^ groups, including mice treated with chemotherapy, no correlation was found between gp130 expression and IL-6 (Figure 8C). These data validate the link between the observed increase in plasma IL-6 level and the expression of gp130 receptor of IL-6 in the Hp, which underlies the decrease in neural stem cell proliferation under chemotherapy, a specific relay compensated by Qiseng^®^.

## 4. Discussion

Chemotherapy-related cognitive impairment (CRCI) and fatigue are common complaints among patients, even at distance of the cancer episode, which contribute to an impact on the QoL of cancer survivors [67]. In particular, a meta-analysis involving 12,327 breast cancer survivors established that patients treated with chemotherapy had a higher risk of developing severe fatigue [68]. Prevention of deleterious effects of chemotherapy on CRCI and/or fatigue is essential, as a significant number of cancer patients are cured. Currently, there is no pharmacological prevention of CRCI, but a significant improvement in fatigue has been observed in breast cancer patients treated with standard treatments receiving intravenous vitamin C during and after 6 months post-injection compared to the control group [19]. In addition, *P. qfolius* extracts administered to a large cancer patient cohort showed a significant reduction in fatigue [12,13]. In the present study, we implemented a mouse model treated with the chemotherapy 5-FU and/or receiving Qiseng® or vitamin C to study spontaneous activity and emotional reactivity. We established that 5-FU caused a significant decrease in locomotor and exploratory activity involving motivation, behaviors likely attributed to fatigue [69], and in cognitive functions. The 5-FU-relayed mechanisms involved intestinal dysbiosis and inflammation, systemic plasma cytokines, brain neuroinflammation and altered Hp neurogenesis. We showed that vitamin C alone administered *per os* tended to attenuate some of the 5-FU effects on cytokines and inflammation, while Qiseng^®^ exhibited a protective role against 5-FU-induced behavioral symptoms resembling neurofatigue, as well as activation of a gut–brain inflammatory axis.

Weight loss is common in chemotherapy patients, contributing to a reduced QoL [70]. Here, repeated administrations of 5-FU evoked weight loss increasingly difficult to recover after each injection. These specific symptoms were already described in non-bearing cancer animals receiving 5-FU chemotherapy [49,61]. In mice, a subcutaneous administration of vitamin C failed to prevent cisplatin-induced weight loss [71], but a *per os* pre-treatment with red ginseng improved the reduction in body weight following 5-FU in rats [72]. Here, neither *per os* vitamin C nor Qiseng^®^ when administered after the first injection of 5-FU prevented or counteracted weight loss induced by chemotherapy. These weight losses were found, for a 5-FU dosage of 50 or 100 mg/kg per day for five days, associated with low mortality in mice [73]. We observed that 5-FU administered at 60 mg/kg caused mortality in the placebo (11.5%) and vitamin C (16%) groups, a percentage likely reduced (4%) but not statistically relevant in the Qiseng^®^ group. The 60 mg/kg dose was optimal to induce intestinal mucositis in mice [73] without reporting mortality [74,75]. It can be hypothesized that the cytotoxic effect of chemotherapy, combined with the stress induced by handling, i.p. injections and gavage and daily gavage procedure is responsible for inflammatory lesions of the esophagus, can contribute to the observed mortality [76,77]. Considering that *P. qfolius* has been shown to inhibit NOX2 and TNF-α expression in the heart of mice exposed to lipopolysaccharide (LPS), thus improving survival [78], we suggest that Qiseng may exhibit protective effect against the forced-feeding-induced lesions and of the pro-oxidative mechanisms of 5-FU [20].

In animals, fatigue can be assessed by evaluating a panel of behaviors, but it must be conceded that the daily activity [69] was chosen as a criterion in the mouse models of chemotherapy-induced fatigue. Indeed, a reduction in locomotor activity in OFT observed in mice treated with 5-FU was reported as fatigue [75]. In our study, mice with 5-FU showed a significant reduction in locomotor activity in OFT only during the awake phase, without shifting the circadian rhythm. In addition, 5-FU mice traveled less distance in the center of the OFT and exhibited less exploration of a new and aversive environment in the LDB test, which may indicate an increase in thigmotaxis and/or anxiety-related behavior, as previously reported [29]. Accordingly, rats treated with doxorubicin and/or cyclophosphamide also exhibited a lower number of entries and time spent in the light compartment than untreated rats [53]. This aversion to mild stressors was not related to a depressive-like behavior, since we did not detect statistical differences between the three groups of 5-FU in TST, FST and sucrose preference test (anhedonia). This is in agreement with the study of Dubois et al., showing no impact of 5-FU in mice on depressive-like behavior [49]. Together, we propose that chemotherapy-induced symptoms of diminished spontaneous, locomotor activity and less motivation to exploration in an aversive area could indicateneurofatigue. In 5-FU mice receiving Qiseng^®^, we detected neither alteration of locomotor activity nor of exploration behavior, even in an aversive environment. These results are in agreement with the compensatory effect of *P. qfolius* on the decreased locomotor activity and anxiety-like behavior induced by sleep deprivation in mice [79]. Together, 5-FU-induced fatigue can be prevented by Qiseng, even if administered after starting the treatment course.

Stress responses and anxiety but also behaviors associated with motivation and activity (fatigue) have been associated with the vHp [80]. Here, mice receiving chemotherapy exhibited less motivation for exploration of an aversive environment and showed a decrease in the time spent in the center of the open field and in the light zone of the LDB associated with a reduction in neurogenesis in the ventral gyrus. These results are consistent with the deleterious effects of 5-FU and leucovorin on the survival of dividing cells in the subgranular area of the dentate gyrus, lasting several weeks in rats [55]. Thus, according to a link between fatigue and vHp, c-Fos increased expression in the dentate gyrus of the vHp was detected following sleep deprivation in rats, suggesting the activation of vHp in fatigue recovery [81]. Interestingly, chemogenetic stimulation of neurogenesis in the ventral dentate gyrus decreases the activity of stress-responsive cells during exploration of an anxiogenic environment, such as the center zone of the OFT [82]. Here, when chemotherapy mice were treated with Qiseng^®^, there was no alteration of the locomotor and anxiety-like behaviors and of neurogenesis in the vHp compared with the NaCl and vitamin C groups. We can conclude that fatigue induced by chemotherapy is linked to a proliferation decrease in the neural precursors, specifically at the level of the vHp, either by a direct anti-mitotic action or *via* neuroinflammation, a cell impact specifically compensated by Qiseng^®^. 

Many studies primarily investigated the cognitive impact of chemotherapy and 5-FU in particular; thus, links have been made between the alteration of cell proliferation and neurogenesis in the dentate gyrus, decreased cognitive performance [83], while a reduction in dendritic spines and an increase in the inflammatory cytokines IL-17, IL-1β and GMCSF in the hippocampus have also been described [84]. More specifically, in mouse dHp, it was shown that a decreased neurogenesis strongly impacted the cognitive processes solicited during the MWM test, supporting the key role of dHp in cognition, learning and spatial memorization [56]. We also evidenced that the chemotherapy mice exhibited altered spatial learning and memory, long-term memory and behavioral flexibility. Analyses of the brain slices from the chemotherapy mice revealed a marked reduction in neurogenesis in the dHP, and even if Qiseng^®^ favored neural cell proliferation, neither neurogenesis in the dorsal gyrus nor the cognitive functions were improved. These observations indicate that repeated administrations of 5-FU impact both fatigue via the vHp neurogenic processes and cognitive function via the dHp neurogenic mechanisms. The selective protective effect of Qiseng strongly supported the distinct chemotherapy mechanisms on fatigue and cognition. In fact, the dorsal dentate gyrus contained more immature neurons (doublecortin+) than the ventral dentate gyrus while being composed of more mature neurons (NeuN+) [56], and the maturation of adult-born neurons differed between the ventral and the dorsal dentate gyrus [85], explaining differential susceptibilities to cytotoxic drugs. In addition, studies showing that i.p. LPS or traumatic brain injury induces acute and/or sustained cytokines mRNA expression in the dHp and vHp, respectively [86,87], suggest that the dHp develops this neuroinflammation first and transiently, which then spreads to the vHp over a longer time. While 5-FU administration drastically altered neurogenesis in the dHp and VHp, Qiseng^®^, and potentially its components, such as ginsenosides or *P. qfolius* metabolites, would be more specific to the neuroinflammatory pathway recruited in the vHp. 

The specific impact of Qiseng^®^ administered *per os* on fatigue should be closely linked to the primary impact of chemotherapy with intestinal inflammation. Chemotherapy adversely affects the gut microbiota [88], and a modification of the intestinal microbiota plays an important role in the development of inflammatory intestinal diseases [59]. It is known that intestinal microbiota plays a fundamental role in the structuring and functioning of the brain [89] and that dysbiosis of the gut microbiota would control the brain functions through a gut–brain axis involving the immune system, the ascending vagal nerve pathway or plasma metabolites and neurotransmitters [90]. At the phylum level, 5-FU treatment has been shown to increase Verrucomicrobia and decrease Firmicutes in BALB/c mouse cecum [61]. Contrary to the results of this paper, in the present study, 5-FU did not impact Firmicutes, and a decrease in Verrucomicrobia was only identified in the 5-FU/Qiseng^®^ group.

A human study interested in Panax ginseng metabolism revealed that a low level of Proteobacteria in feces was associated with high metabolism of Rb1 ginsenosides present in ginseng to compound K [91]. Indeed, some ginsenosides such as Rb1 and Rc, were digested by gastric acid and/or intestinal bacteria into smaller compounds, such as Rd and compound K, and further into Protopanaxadiol [58]. In our chemotherapy mouse model, we established for the first time the role of 5-FU in increasing Lactobacillus, and more specifically, L. reuteri, L. acidophilus and L. johnsonii/gasseri, and decreasing L. murinus/animalis. This chemotherapy-induced elevation of Lactobacillus species dysbiosis was mainly prevented by Qiseng^®^ but also vitamin C, but concomitantly, in all Qiseng^®^ mice, a specific diminution of Beta-, Gamma- and Delta-proteobacteria was observed. Thus, Qiseng^®^ modified the gut microbiota equilibrium through a decrease in proteobacteria while preventing the increase in Lactobacillus induced by 5-FU. Similarly, ginsenoside supplementation can restore the composition of the microbiota altered by chemotherapy [92]. Additionally, it was shown that a decoction of Ginseng could restore D-galactose-induced memory deficits in rats by up-regulating Bacteroidetes and down-regulating Lactobacillus [93].

These preclinical data are in agreement with a randomized clinical trial concluding that probiotic supplementation during chemotherapy could reduce the incidence of CRCI, by means of an increase in the plasma metabolite p-Mentha-1,8-dien-7-ol and potentially reduced inflammation in breast cancer patients [94],. Although several Lactobacillus species are currently used as probiotics to improve gastrointestinal health, increased levels of Lactobacilli are not always associated with healthy conditions. For example, higher levels of Lactobacilli were reported in obese individuals [95], as well as in mouse models of undernutrition [96]. Lactobacilli may, in some instances, activate the immune system by engaging host pattern-recognition receptors and trigger inflammation [97], playing a role in bowel inflammation disease [93]. Indeed, the oral administration of L. reuteri to Lactobacillus-free mice resulted in an inflammatory reaction of the intestinal mucosa 6 days after inoculation, correlated with IL-1α and IL-6 mRNA expression in small intestine enterocytes [98]. Moreover, L. acidophilus can promote cell reactive oxygen species (ROS) in cultured intestinal epithelial cells [99], whose production likely contributes to intestinal tissue damage and triggers NF-κB, leading to pro-inflammatory cytokines, such as TNF-α, IL-1β and IL-6 [16,17]. Thus Qiseng^®^ but also vitamin C may control intestinal mucosa inflammation during chemotherapy. 

Here, we evaluated the serum levels of ginsenosides in the control and 5-FU mice after two weeks of Qiseng^®^ administration and mainly found Rb1, Rd and Rc, and also the compound K metabolite [20], Protopanaxadiol. The absence of compound K can be explained by a rapid metabolization. Indeed, in rats, it was shown that compound K was detectable in blood from 4 h to 7 h post-oral-administration of Rb1 [100]. Since we collected blood samples at least 24 h post-Qiseng^®^ oral treatment, the absence of compound K can be correlated to its metabolization into Protopanaxadiol, here detected in some samples. The levels of ginsenosides found after 24 h are, nevertheless, in agreement with previous studies showing a proportion of Rb1 and Rc ginsenosides unmetabolized by bacteria after their administration [100,101]. In our study, Rc was even detected in higher concentrations in the serum of 5-FU mice, suggesting less metabolization, possibly due to the chemotherapy-induced dysbiosis. Interestingly, the ginsenoside Rc would possess the highest antioxidant activity, followed by Rb1, then Rd, preventing the formation of ROS, whose elevation is involved in inflammatory processes and production of pro-inflammatory cytokines [102]. Thus, Qiseng^®^-derived ginsenosides should contribute to reducing the 5-FU impact on intestinal mucosa and the intestinal wall, while the ginsenosides available in the bloodstream may, in addition, play neuroprotective roles. 

Intestinal damages following 5-FU treatment were characterized here by a decrease in the intestinal surface and villi height, correlated with an increase in intestinal inflammation corroborated by the infiltration of MOMA+ macrophages. In complete agreement, a decrease in intestinal villi height and increased inflammatory cell infiltration in both mucosal and sub-mucosal intestines have been attributed to chemotherapy [103]. Interestingly, the American ginseng polysaccharide and ginsenosides can relieve cyclophosphamide-induced damage by protecting mucosal surfaces against infection [92]. Our results showed that Qiseng^®^ only partially protected the intestinal surface, while the 5-FU-induced diminished height of the villi was not significantly prevented. However, Qiseng^®^ compared to vitamin C lowered monocyte/macrophage infiltration caused by chemotherapy. Intestine-activated macrophages may produce pro-inflammatory cytokines, including TNF-α, IL-1β, IL-6 but also ROS [104]. In particular, IL-6 has been shown to increase the epithelium tight junction permeability of Caco-2 monolayers [105], potentially responsible for bloodstream diffusion of microbial products triggering systemic inflammation and increased serum IL-6 level [106]. 

Changes in the composition of the cecal microbiota and intestinal inflammation have been associated with elevation of pro-inflammatory cytokines [59]. In a mouse model of colonic mucositis induced by 5-FU, the increased serum levels of IL-6 and TNF-α were already described [61]. In the present study, 5-FU also evoked plasma levels of IL-6 and TNF-α as well as IL-2, IL-4, IL-12p70, IL-17 and MCP-1 while decreasing RANTES. We demonstrated positive correlations between enhanced IL-6 and MCP-1 with increased proportion of Lactobacilli, and more specifically, L. acidophilus. Interestingly, L. acidophilus could upregulate a number of effector genes, such as IL-6 and CCL2 (MCP-1) in mice, involving NF-κB and p38 Mitogen-Activated Protein Kinase (MAPK) signaling pathways in intestinal epithelial cell lines [65]. We also observed that the co-administration of vitamin C with 5-FU suppressed plasma IL-2, IL-4, IL-12p70, IL-17 and TNF-α cytokines but did not prevent increased IL-6 and MCP-1 as well as decreased RANTES levels. Such data closely corroborate a partial but not complete prevention of IL-6 expression by vitamin C supplementation in a mouse model of colon cancer [107]. A remarkable effect of Qiseng^®^ was thus detected, since all the measured cytokines were found at the level of the NaCl mice, possibly due to the Rb1, Rd and Protopanaxadiol ginsenosides resulting from Qiseng^®^ metabolization. Indeed, the administration of Rb1 for 23 days in a cancer-induced cachexia mouse model led to diminished TNF-α and IL-6 levels in serum evoked by cancer [26]. Similarly, Rd was able to alleviate the colitis induced by E. Coli, in particular by suppressing the increased expression of IL-6, thus normalizing the anxiety/depression behaviors [108], suggesting obvious links between dysbiosis, intestinal inflammation, circulating cytokines, and more specifically, IL-6.

In cancer patients, cytokine circulating levels, particularly IL-6, have been positively correlated with scores of cognition and somatic fatigue [14,62]. In mice, a single intravenous injection of IL-6 produced an increased immobility time [63] and a significant decrease in locomotor activity [15]. We demonstrated that 5-FU mice receiving placebo or vitamin C and exhibiting at least an increased plasma IL-6 level show marked decrease in locomotor activity and exploration behavior. Because the anxiety-like (EPM test) and depressive-like (TST/FST) behaviors were not significantly modified when 5-FU was administered, we propose that 5-FU evoked symptoms of neurofatigue associated, at least in part, with IL-6. Qiseng^®^ supplementation during chemotherapy notably extinguished IL-6 in serum and rehabilitated mouse locomotor activity and exploration. While BBB can selectively transport several circulating cytokines, such as TNF-α or IL-6, leading to neuroinflammation [109], it should be noticed that IL-6 tg^+/+^ mice with a high level of IL-6 expressed by astrocytes presented a reduced activity/exploration in OFT and LDB compared with IL-6 tg^−/−^ mice [110]. Some studies already reported a role of the glycoprotein 130 (gp130) receptor for IL-6 in astrocytes but also in neurons. For example, in a mouse model of autoimmune encephalomyelitis, mice with astrocytes expressing the gp130 receptor were shown to develop astrogliosis [111]. In our study, we observed in chemotherapy mice a net immunolabeling of gp130 detected both in mature neurons but also in astrocytes within the Hp correlated with decreased neural precursor cell proliferation in the dentate gyrus. Similar results indicate that overexpression of IL-6 by astrocytes decreased neurogenesis by 63% in the Hp dentate gyrus of adult mice [112]. Here, Qiseng^®^ administered with chemotherapy blocked serum IL-6 and attenuated the expression of IL-6 gp130 receptor in the dHp and vHp through mechanisms potentially involving ginsenosides. This control of 5-FU-induced neuroinflammatory processes is likely more specifically associated with diminished neurogenesis in the vHp and alteration of locomotor activity.

We recognize, however, that this work has several limits. First, we are fully aware that we interpreted “fatigue” in animals through the association of exploratory and motivation behaviors and spontaneous activity and that we cannot claim that these behaviors reflect fatigue. Among patients, cancer-related fatigue (CRF), as defined by the National Comprehensive Cancer Network (NCCN), is “a distressing, persistent, subjective sense of physical, emotional and/or cognitive tiredness or exhaustion related to cancer or cancer treatment that is not proportional to recent activity and interferes with usual functioning” [113]. Interestingly, it has been shown in humans that decreased motivation and fatigue are both related to alterations in the basal ganglia and reduced neuronal activation of the reward signals [114,115], suggesting similar neurological substrates. Thus, we and others focused on cancer-related fatigue in animals related to tumor growth [116], chemotherapy [117] or radiation [118] by using a common physical activity criterion by means of OFT [119] also involving motivation. Second, this work was exclusively conducted on C57Bl/6 male mice to avoid behavioral issues due to the estrous cycle in female mice. Accordingly, it was shown in female mice that the estrus cycle has a robust effect on anxiety, low anxiety being associated with a high estradiol level (proestrus and estrus phase) [120]. Nevertheless, a very important study in young mice exposed to a combination of chemotherapy established that males, and not females, exhibited deficits in short memory and executive functions 4–5 weeks after chemotherapy [121]. Interestingly, this early exposure to chemotherapy led to an increased MCP-1 expression only in males, while the oligodencrocyte markers were altered in both males and females. Thus, the present work should be reproduced in female mice.

## 5. Conclusions

Advances in diagnostic and therapeutic strategies in oncology have significantly increased the chance of survival of cancer patients, even those with metastatic disease. To better understand the pathophysiology and neurobiological mechanisms of CRCI and the direct impact of the different cancer treatments, animal models have been developed to investigate the selective effects of the disease and/or treatment on the neurocognitive function and the influence of stress, mood and aging on cognitive impairment. It must be said that growing demands for CRCI management from patients led to studies testing cognitive rehabilitation in cancer patients but that animal models should provide cues of efficacy of pharmacological and non-pharmacological care and support of CRCI and fatigue.

Here, we proved that frequent administrations of 5-FU, in the absence of cancer, induced the loss of locomotor/exploratory activity that we associated with neurofatigue, coupled with cognitive deficits (spatial learning and memory and flexibility). In these 5-FU mice, the gut microbiota composition was altered and the intestinal mucosa was inflamed, both associated with proinflammatory serum cytokines, notably IL-6. Vitamin C did not compensate for the effects of 5-FU on activity/fatigue, alteration of the microbiota and elevation of IL-6. 

A *P. qfolius*-based solution (Qiseng^®^) completely prevented the deleterious effect of 5-FU on locomotor and exploration activity while preventing microbiota dysbiosis and elevation of all cytokines. This preclinical study established that Qiseng^®^ supplementation could prevent chemotherapy-induced neurofatigue but not cognitive impairment. These data provide an important validation of the beneficial effect of a Qiseng^®^ supplementation on fatigue in cancer patients treated with chemotherapy, while this type of non-pharmacologic intervention should be more accepted by patients. Accordingly, a recent clinical trial “Evaluation of the Impact of Taking American Ginseng for 8 weeks on Fatigue in Patients Treated for Localized Breast Cancer (QISEIN)” (NCT05241405) was initiated very recently, with the objective of improving cancer patients’ QoL. We think that the Qiseng^®^-induced benefit is not trivial, since mental fatigue affects large-scale network connectivity in the brain during high cognitive demands associated with effort [122]. In fact, preventing neurofatigue should enable optimal cognitive functions or acceptance of the effort required in a cognitive rehabilitation protocol.

## Figures and Tables

**Figure 1 cancers-14-04403-f001:**
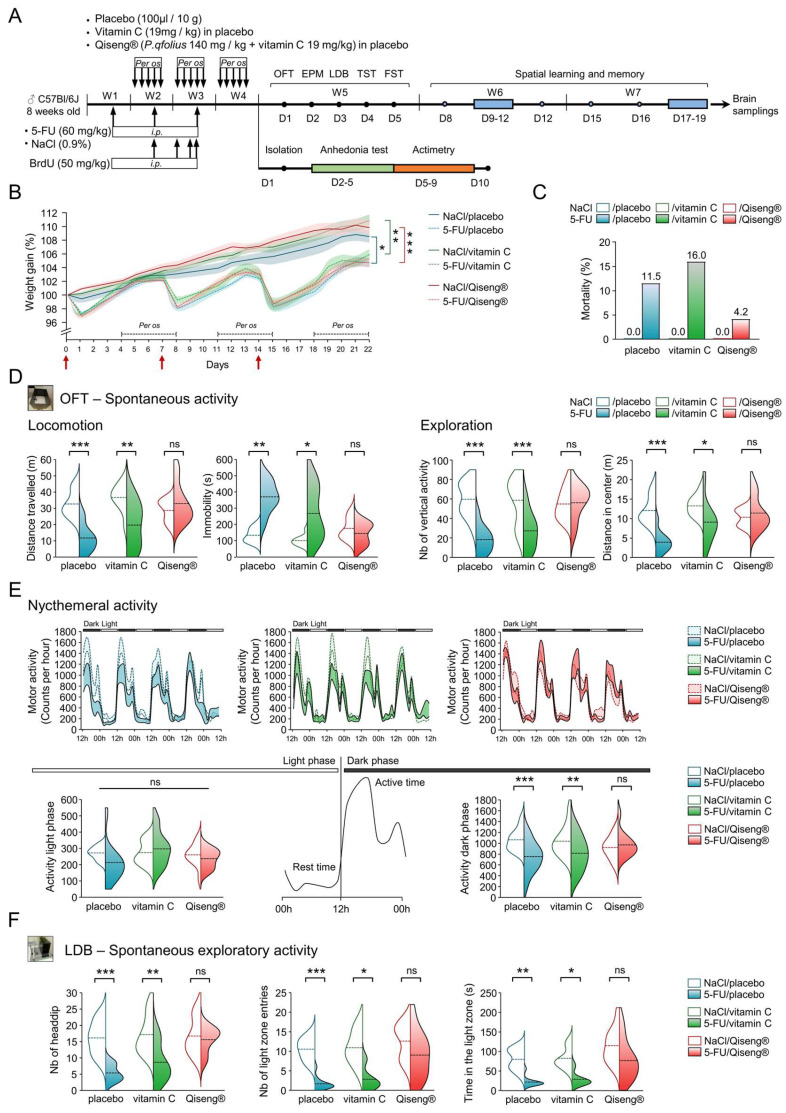
Preventive impact of Qiseng^®^ on the behavioral consequences of 5-FU for spontaneous activity and exploration and nycthemeral rhythms and activity. (**A**) Schematic representation of the schedule and experimental protocol showing the timeline for the different treatments and behavior tests assessing emotional reactivity and cognitive functions in mice. C57BL/6J Rj mice received intraperitoneal injection of 5-FU (60 mg/kg, red arrow) once a week for 3 weeks (i.p., *arrow*) and/or per os administration of Placebo, vitamin C (19 mg/kg) in placebo or Qiseng^®^ (*P. qfolius* 140 mg/kg + vitamin C 19 mg/kg) once a day, 5 times a week for 3 weeks. Mice also received consecutive BrdU injections (50 mg/kg, once a day for 4 days, *arrow*) in weeks 2 and 3. Behavioral evaluations started at week 4 (W4), with the first group of mice evaluated in the open field test (OFT) on D1, elevated plus maze (EPM) on D2, light/dark box test (LDB) on D3, tail suspension test (TST) on D4 and forced swim test (FST) on D5. Cognitive functions were investigated at W6 and W7, and mouse brains were collected at D19. For a second series of experiments, mice were individually housed from D1 of W4 and evaluated in the sucrose preference test (D2 to D5) followed by testing in actimetry (D5 to D9). (**B**) Curves of body weight gain during the different treatment periods. NaCl/placebo (*n* = 22), 5-FU/placebo (*n* = 24), NaCl/vitamin C (*n* = 23), 5-FU/vitamin C (*n* = 23), NaCl/Qiseng^®^ (*n* = 21), 5-FU/Qiseng^®^ (*n* = 24). Bi-directional repeated-measures ANOVA; * *p* < 0.05, ** *p* < 0.01, *** *p* < 0.001. (**C**) Percentage of mortality occurring over the course of the 5-FU injection in the absence or the presence of placebo, vitamin C or Qiseng^®^. (**D**) Impact of 5-FU in the absence or the presence of vitamin C or Qiseng^®^ on spontaneous activity and exploration in OFT (**D**), nycthemeral rhythms, light and dark phase activity (**E**) or spontaneous exploratory activity in the LDB (**F**). 5-FU/placebo and 5-FU/vitamin C mice showed reduced spontaneous activity, locomotor exploration and dark phase activity compared with respective NaCl mice. Data distributions are shown in violin plots, with the median in dotted line. NaCl/placebo (D/F: *n* = 10, E: *n* = 12), 5-FU/placebo (D/E/F: *n* = 12), NaCl/vitamin C (D/F: *n* = 10, E: *n* = 12), 5-FU/vitamin C (D/E/F: *n* = 11), NaCl/Qiseng^®^ (D/E/F: *n* = 10), 5-FU/Qiseng^®^ (D/E/F: *n* = 12). Kruskal–Wallis-Dunn post hoc; * *p* < 0.05, ** *p* < 0.01, *** *p* < 0.001, ns: not significant. 5-FU: 5-fluorouracil, BrdU: 5-bromo-2’-deoxyuridine, D: Day, W: Week, i.p.: intraperitoneal injection, OFT: open field test, EPM: elevated plus maze, LDB: light/dark box, TST: tail suspension test, FST: forced swim test.

**Figure 2 cancers-14-04403-f002:**
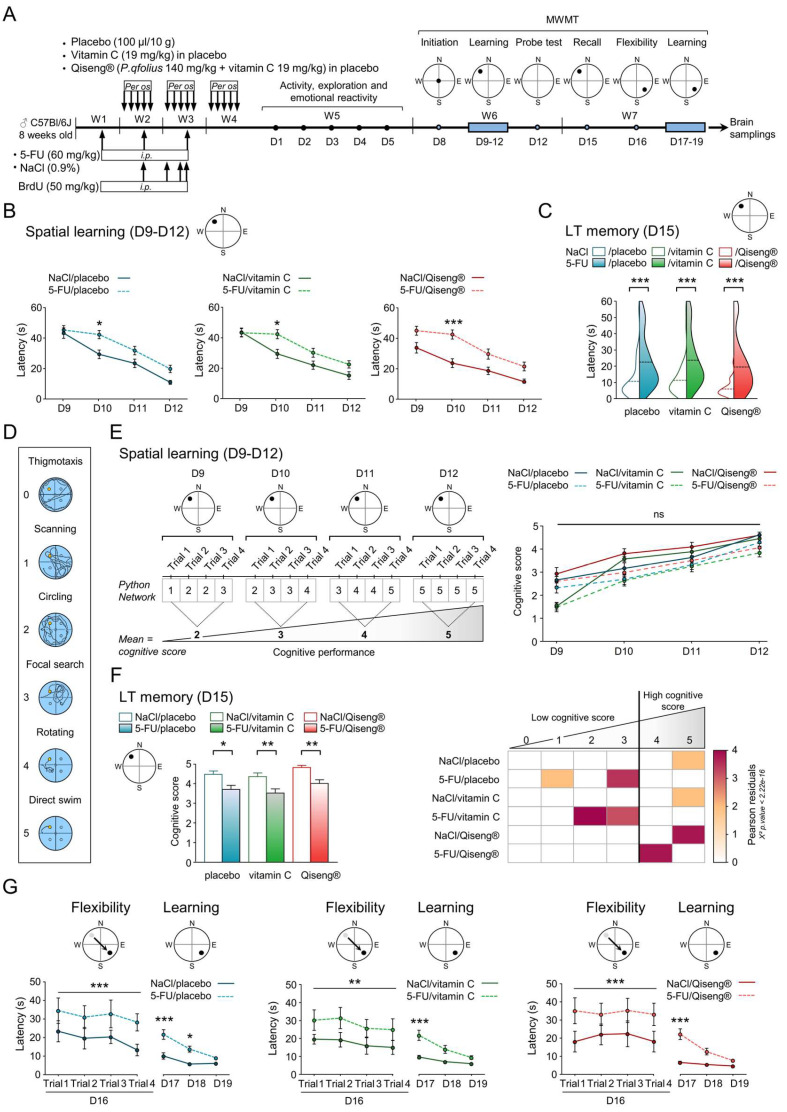
Absence of preventive effect of Qiseng^®^ on the 5-FU-induced alteration of spatial learning, memory and behavioral flexibility. (**A**) Schematic diagram showing the timeline for treatments and behavioral experiments assessing activity, exploration and emotional reactivity already described in Figure 1 but also the cognitive functions evaluated in the MWMT for the same groups of mice. Mice received one i.p. injection of 5-FU once a week for 3 weeks (*arrow*), while placebo solution, vitamin C (19 mg/kg) in placebo or Qiseng^®^ (*P. qfolius* 140 mg/kg + vitamin C 19 mg/kg) were administered per os once a day, 5 times a week for 3 weeks. Mice also received i.p. BrdU injections (50 mg/kg, once a day for 4 days, *arrow*) in W2 and W3. MWMT was performed over 10 days, starting from W5 with the familiarization on D8, the first learning phase from D9 to D12, the probe test on D12, two hours after learning trials, the retrieval test on D15 and the behavioral flexibility analyzed on D16, followed by the second learning phase from D17 to D19. Animal brains were then collected at the end of D19. (**B**) Cognitive consequences of 5-FU administration in the absence or the presence of vitamin C or Qiseng for spatial learning location (*n* = 10–12, bi-directional repeated-measures ANOVA; * *p* < 0.05, ** *p* < 0.01, *** *p* < 0.001). (**C**) Impact of 5-FU on long-term memory (LT memory) in the absence or the presence of vitamin C or Qiseng^®^. Data distributions are shown in violin plots, with the median in dotted line (*n* = 10–12, Kruskal–Wallis-Dunn post hoc; *** *p* < 0.001). (**D**) Illustration and classification with an attributed scoring of swimming strategies to localize the platform in the spatial learning and memory, from the lowest to the highest: Thigmotaxis (0) < Scanning (1) < Circling (2) < Focal search (3) < Rotating (4) < Direct swim (5). (**E**) On the left, schematic representation of platform positioning for each day of the spatial learning phase (D9–D12) of the MWMT: four trials with four different start locations (N, S, W, E) were investigated each day. The Python software (see Materials and Methods) determined the strategy used by the animal to reach the platform in each trial and assigned a score according to the scale described in D (*n* = 10–12, bi-directional repeated-measures ANOVA; ns: not significant). (**F**) Effect of 5-FU on the cognitive score during the recall phase (D15). Bars are mean ± SEM (*n* = 10–12, Kruskal–Wallis-Dunn post hoc; * *p* < 0.05, ** *p* < 0.01). On the right, a χ^2^ test associated with Pearson’s residual test was proposed to extract significant variations in the swimming strategies used by the different groups to localize the platform. The higher the value of Pearson’s residual test, the more frequent is the distribution of score event was when compared to the other groups. Strategies were ranked as in D, from the least to the most efficient ones, to locate the platform. (**G**) Effect of 5-FU on spatial learning and memory or behavioral flexibility. In the flexibility phase, the position of the platform in the NW quadrant during the learning and recall tests (D9 to D15) was changed to the SE quadrant (D16 to D19). All mice treated with chemotherapy compared with NaCl-treated mice showed an increased latency to find the platform location at D16 (flexibility) during the four trials, also detected at D17, while they learned the new location of the platform from D18 to D19 reaching the level of all NaCl mice. In each analysis, NaCl/placebo (*n* = 10), 5-FU/placebo (*n* = 12), NaCl/vitamin C (*n* = 10), 5-FU/vitamin C (*n* = 11), NaCl/Qiseng^®^ (*n* = 10), 5-FU/Qiseng^®^ (*n* = 12). Bi-directional repeated-measures ANOVA; * *p* < 0.05, ** *p* < 0.01, *** *p* < 0.001). 5-FU: 5-fluorouracil, BrdU: 5-bromo-2’-deoxyuridine, D: day, MWMT: Morris water maze test, LT: Long term.

**Figure 3 cancers-14-04403-f003:**
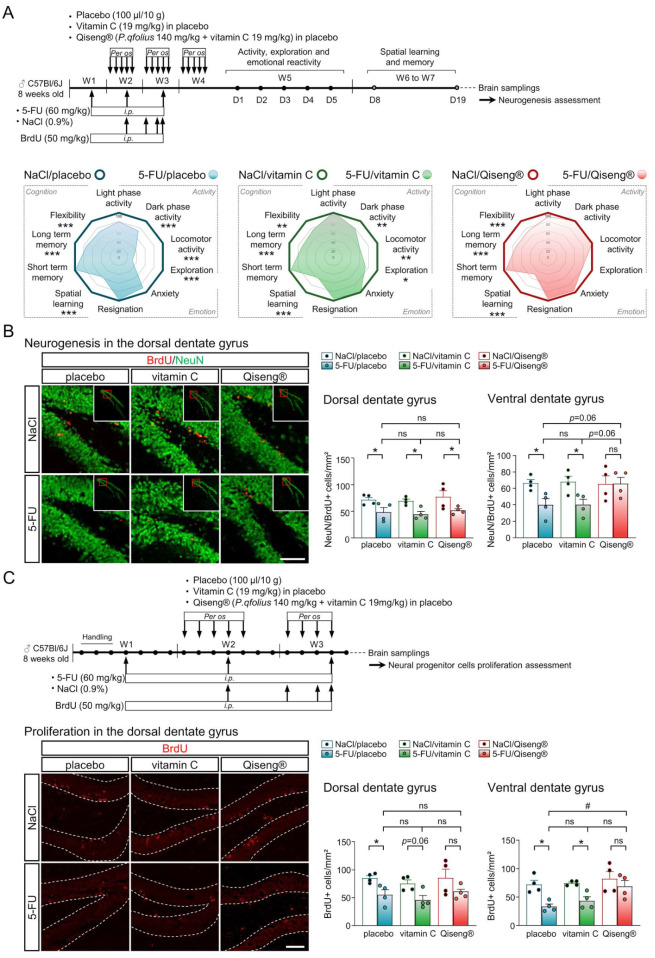
Preventive effect of Qiseng^®^ on 5-FU-induced altered activity and exploration and decreased precursor proliferation and neurogenesis in ventral and dorsal hippocampus. (**A**) Schematic representation of the schedule and experimental protocol showing the timeline for treatments and behavior experiments (see Figure 1 and Figure 2). C57BL/6J Rj mice received one injection of 5-FU once a week for 3 weeks (i.p., *arrow*) and *per os* administration of placebo solution, vitamin C (19 mg/kg) in placebo or Qiseng^®^ (*P. qfolius* 140 mg/kg + vitamin C 19 mg/kg) once a day, 5 times a week for 3 weeks. Mice also received i.p. BrdU injections (50 mg/kg, once a day for 4 days, arrow) at W2 and W3. Behavioral evaluations started at W5 to W7. Mouse brains were collected at the end of D19 to evaluate the impact of treatments on neurogenesis in Hp. Radar plots illustrating the impact of all treatments on performance in the activity, emotional and cognitive domains. Each item of the radar represents the mean normalized to the respective NaCl groups, NaCl/placebo (*n* = 10–11), 5-FU/placebo (*n* = 12), NaCl/vitamin C (*n* = 10–12), 5-FU/vitamin C (*n* = 11), NaCl/Qiseng^®^ (*n* = 10), 5-FU/Qiseng^®^ (*n* = 12). * *p* < 0.05, ** *p* < 0.01, *** *p* < 0.001. (**B**) Representative microphotography of BrdU-(red) and NeuN-(green) positive cells in dHp and vHP from groups studied in Figure 1 and Figure 2. Neurogenesis was evaluated as the number of BrdU+/NeuN^+^ cells in dHp and vHp. Bars are mean ± SEM with symbols of individual data points. NaCl/placebo (*n* = 4), 5-FU/placebo (*n* = 4), NaCl/vitamin C (*n* = 4), 5-FU/vitamin C (*n* = 4), NaCl/Qiseng^®^ (*n* = 4), 5-FU/Qiseng^®^ (*n* = 4). Kruskal–Wallis-Dunn post hoc; * *p* < 0.05, ns: not significant). Scale bar: 100 μm. (**C**) Schematic diagram showing the timeline for treatments. Mice received an i.p. injection of 5-FU once a week for 3 weeks (*arrow*) and *per os* administration of placebo, vitamin C (19 mg/kg) in placebo or Qiseng (P. qfolius 140 mg/kg + vitamin C 19 mg/kg) once a day, 5 times a week for 2 weeks. Mice also received four consecutive i.p. BrdU injections (50 mg/kg, *arrow*) at W2 and W3, and brains were collected at the end of the treatment session. Representative microphotography of BrdU-(red) stained positive cells in the hippocampal dentate gyrus. Bar graphs represent quantification of the number of BrdU^+^ cells/mm^2^ in the dentate gyrus of the dorsal *(Left*) and the ventral *(Right*) Hp. Mean ± SEM with symbols of individual data points. NaCl/placebo (*n* = 4), 5-FU/placebo (*n* = 4), NaCl/vitamin C (*n* = 4), 5-FU/vitamin C (*n* = 4), NaCl/Qiseng^®^ (*n* = 4), 5-FU/Qiseng^®^ (*n* = 4). Kruskal–Wallis-Dunn post hoc; ** p* < 0.05, # *p* < 0.001, ns: not significant). Scale bar: 100 μm. 5-FU: 5-fluorouracil, BrdU: 5-bromo-2′-deoxyuridine, NeuN: Neuronal nuclei antigen.

**Figure 4 cancers-14-04403-f004:**
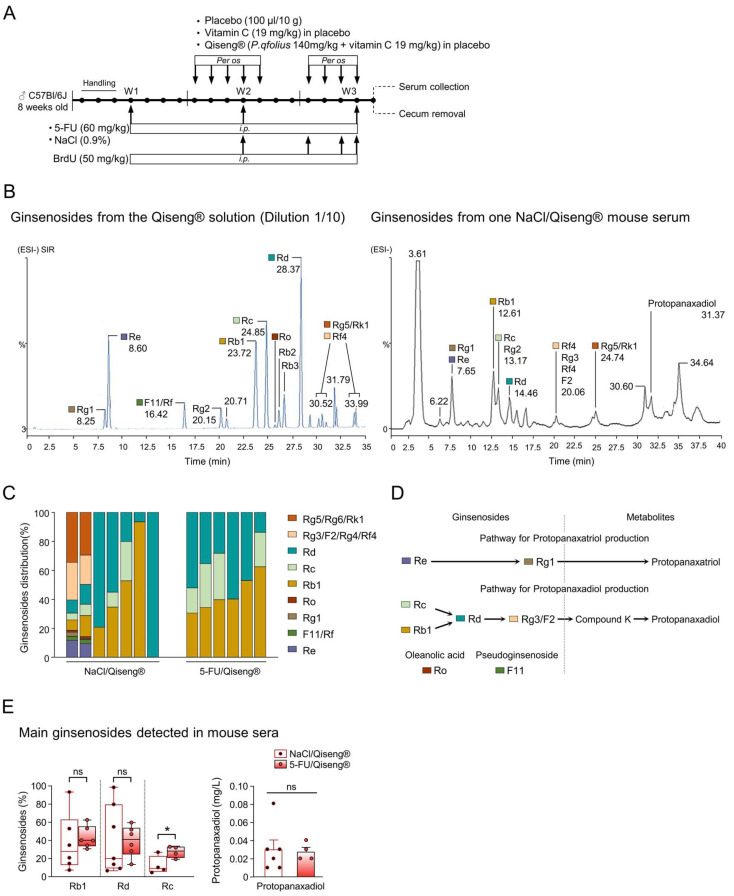
Detection of ginsenosides and protopanaxadiol in sera of placebo and 5-FU mice treated with Qiseng^®^. (**A**) Schematic illustration showing the timeline for treatments and time of serum collection in mice to study the composition in ginsenosides and potential metabolites. C57BL/6J Rj mice received i.p. 5-FU injections once a week for 3 weeks (*arrow*) and *per os* placebo, vitamin C or Qiseng^®^ once a day, 5 times a week for 2 weeks. Mice received four BrdU i.p. injections (once a day) at W2 and W3; then, serum and cecum were collected for analysis. (**B**) HPLC chromatogram and chemoprofile of Qiseng^®^ and some ginsenosides from a 1/10 Qiseng diluted solution (*Left*) and from a representative example of a NaCl/Qiseng^®^-treated mouse serum (*Right*). (**C**) Relative abundance of Rb1, Rc, Rd, Re, Rg5/Rg6/Rk1, Rg3/F2/RG4/Rf4, Ro, Rg1, F11/Rf ginsenosides in sera of NaCl/Qiseng^®^ and 5-FU/Qiseng^®^-treated mice. The Rb1, Rd and Rc ginsenosides were predominantly found in sera of both groups, and the Rc ginsenosides proportion was significantly higher in 5-FU/Qiseng^®^ mice. (**D**) Schematic illustration of the composition of Qiseng^®^ and the potential metabolization by gut microbiota into protopanatriol, protopanaxadiol and Factor K metabolites (*Left*). Sole protopanaxadiol can be detected in only some serum collected from NaCl/Qiseng^®^ and 5-FU/Qiseng^®^ mice. (**E**) Percentage of ginsenosides (*Left*) or Protopanaxadiol (*Right*) in sera of NaCl/Qiseng® and 5-FU/Qiseng® mice. Data are represented as box plots with the median or bar plots of mean ± SEM with symbols of individual data points. In each analysis, NaCl/Qiseng^®^ (*n* = 4–7), 5-FU/Qiseng^®^ (*n* = 4–6). Mann–Whitney; * *p* < 0.05, ns: not significant). 5-FU: 5-fluorouracil, BrdU: 5-bromo-2′-deoxyuridine, D: Day, W: Week, i.p.: intraperitoneal injection.

**Figure 5 cancers-14-04403-f005:**
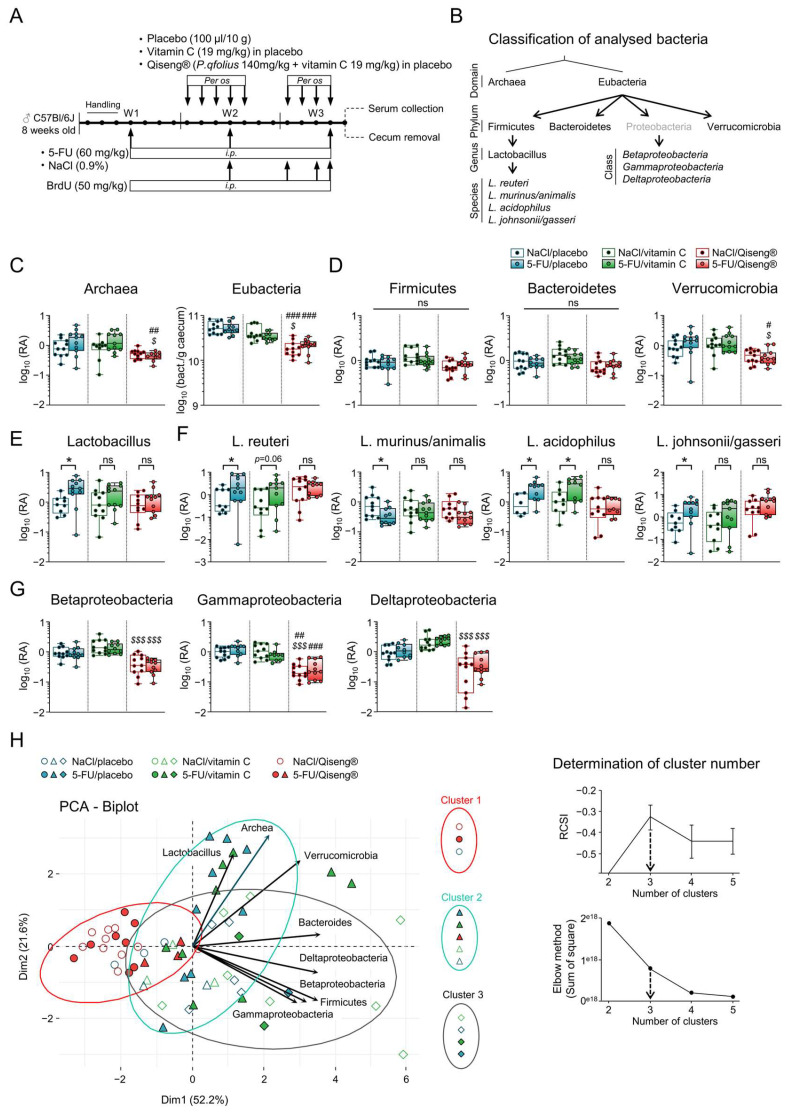
Impact of 5-FU, vitamin C and Qiseng^®^ on the abundance of selected species from the gut microbiota. (**A**) Schematic illustration showing the timeline for treatments and time of serum and cecum collections in mice. (**B**) Schematic illustration showing the classification and relationships between the different domains, phyla, genera, classes and species of gut bacteria quantified by qPCR. (**C**–**G**) Relative abundance (RA) of Archaea and Eubacteria (**C**), Firmicutes, Bacteroidetes and Verrucomicrobia (**D**), the Lactobacillus genera (**E**), *L. reuteri*, *L. murinus/animalis*, *L. acidophilus* and *L. johnsonii/gasseri* species (**F**) and Betaproteobacteria, Gammaproteobacteria and Deltaproteobacteria (**G**) from cecal contents of 5-FU/placebo, 5-FU/vitamin C and 5-FU/Qiseng^®^ and their respective NaCl control groups. Data are presented as whisker plots with median, minimum/maximum values and symbols of individual data points. Stars indicate statistical differences within the same group, dollars indicate statistical differences within the vitamin C group, and hashes indicate statistical differences within the placebo group (Kruskal–Wallis-Dunn post hoc; * *p* < 0.05, # *p* < 0.05, ## *p* < 0.01, ### *p* < 0.001, $ *p* < 0.05, $$$ *p* < 0.001, ns: not significant. In each analysis, NaCl/placebo (*n* = 6–11), 5-FU/placebo (*n* = 10–11), NaCl/vitamin C (*n* = 9–11), 5-FU/vitamin C (*n* = 9–11), NaCl/Qiseng^®^ (*n* = 10–11), 5-FU/Qiseng^®^ (*n* = 9–11). (**H**) Plot of principal component analysis (PCA) of combined data from the abundance of gut bacterial taxa based on dissimilarity among the different NaCl- and 5-FU-treated groups of mice. Both clustering calculations of the relative cluster stability index (RCSI) and elbow methods provided the best score for three clusters (*Right*). Cluster 1 (red) consisted of NaCl/Qiseng^®^ (all, *n* = 10), 5-FU/Qiseng^®^ (*n* = 8) and NaCl/Placebo (*n* = 3) samples. Cluster 2 (blue/green) included 5-FU/Placebo (*n* = 9), 5-FU/vitamin C (*n* = 9), 5-FU/Qiseng^®^ (*n* = 3), NaCl/vitamin C (*n* = 3) and NaCl/Placebo (*n* = 2) samples. Cluster 3 (gray) was composed of NaCl/vitamin C (*n* = 8), NaCl/Placebo (*n* = 5), 5-FU/vitamin C (*n* = 2) and 5-FU/Placebo (*n* = 1). 5-FU: 5-fluorouracil, L.: Lactobacillus, PCA: principal component analysis, RCSI: relative cluster stability index.

**Figure 6 cancers-14-04403-f006:**
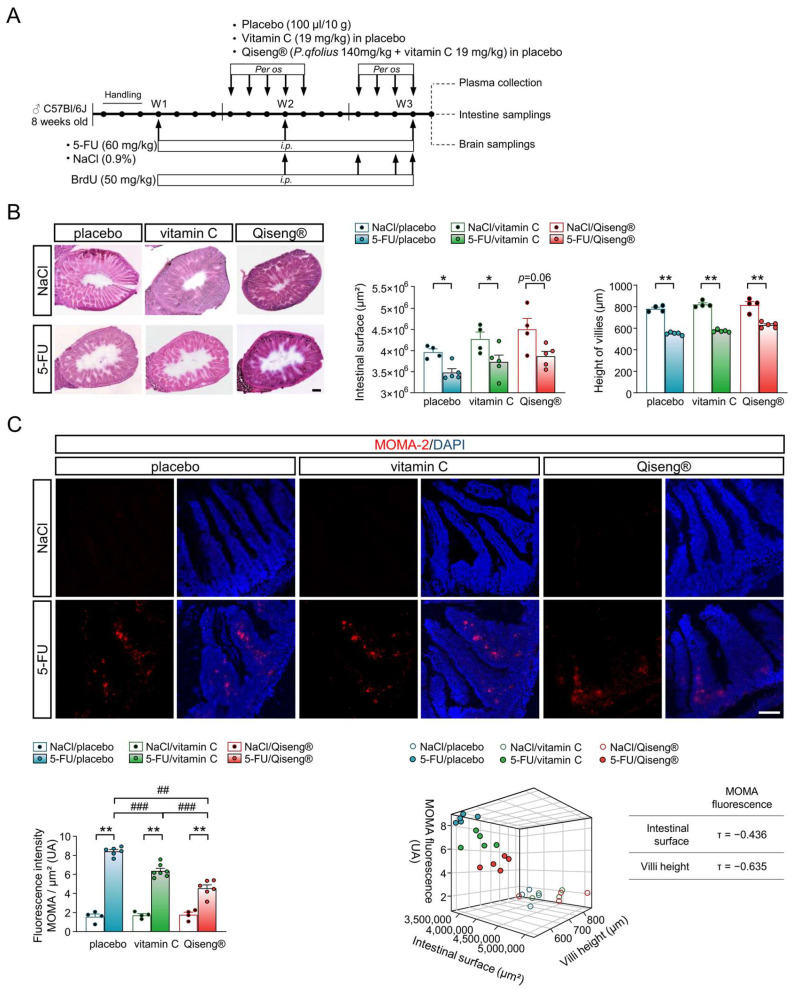
Impact of 5-FU treatment, vitamin C and Qiseng^®^ on intestinal integrity and macrophages infiltration. (**A**) Schematic timeline for treatments and intestine sampling from the different groups of mice for studying small intestinal tissue integrity. (**B**) Representative microphotography of eosin/hematoxylin staining from intestinal transversal sections prepared from the NaCl/ and 5-FU/placebo, vitamin C and Qiseng^®^-treated mice (*Left*) and intestinal surface and villi height quantifications. Scale bar: 200 μm. Data are represented as bar plots of mean ± SEM with symbols of individual data points (*n* = 4–5, Kruskal–Wallis-Dunn post hoc; * *p* < 0.05, ** *p* < 0.01). (**C**) Representative microphotography of the monocyte/macrophage specific marker MOMA-2 (red) in villi of small intestine in NaCl/ and 5-FU/placebo, vitamin C and Qiseng^®^-treated mice. Fluorescence quantification of the effects of 5-FU, vitamin C and Qiseng^®^ on intestinal macrophage infiltration. Bottom left, data are represented as bar plots of mean ± SEM with symbols of individual data points (*n* = 4–5, Kruskal–Wallis-Dunn post hoc; ** or ^##^
*p* < 0.01, ^###^
*p* < 0.001, ns: not significant). Star indicates statistical difference within the same group, and hash indicates statistical difference between 5-FU groups. Scale bar: 100 µm. Bottom right, scatterplot in 3D illustrating the relationships between the MOMA-2 staining, villi height and the intestinal surface from the different treated groups. The negative correlation coefficient of Kendall (^###^) indicates an increase in intestinal inflammation with a decrease in the intestinal surface and height of villi, specifically evidenced in the 5-FU/Placebo group. In each analysis, NaCl/placebo (*n* = 4), 5-FU/placebo (*n* = 5–6), NaCl/vitamin C (*n* = 4), 5-FU/vitamin C (*n* = 5–7), NaCl/Qiseng^®^ (*n* = 4), 5-FU/Qiseng^®^ (*n* = 5–6). 5-FU: 5-fluorouracil, BrdU: 5-bromo-2’-deoxyuridine, D: Day, W: Week, i.p.: intraperitoneal injection, DAPI: 4′,6-diamidino-2-phenylindole, MOMA2: monocytes and macrophages marker.

**Figure 7 cancers-14-04403-f007:**
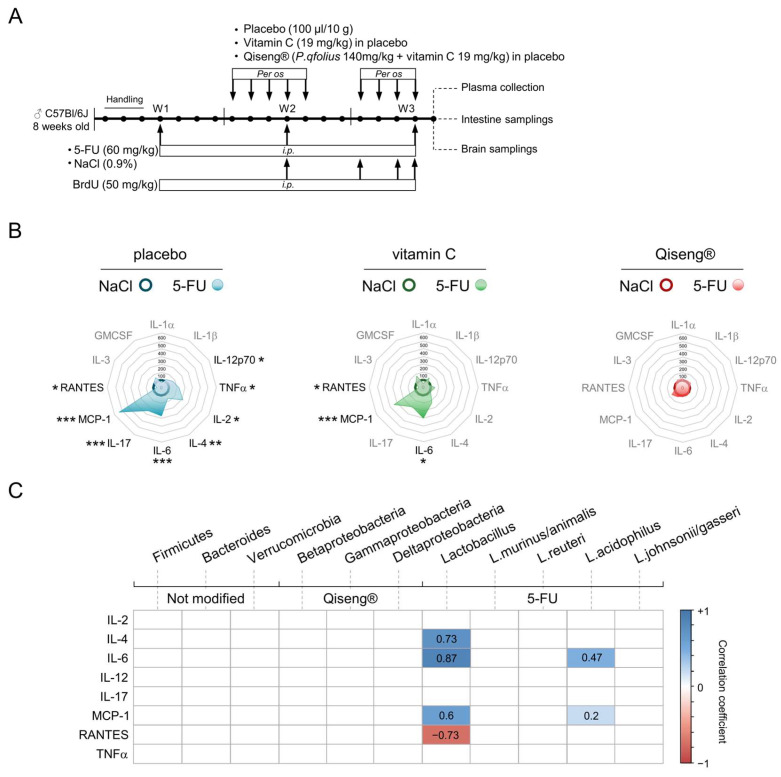
Impact of 5-FU in the absence or the presence of placebo, vitamin C or Qiseng^®^ on plasma levels of pro-inflammatory, pluripotent, chemotactic and leukocyte growth cytokines. (**A**) Schematic timeline of treatments and intestine sampling from the different groups of mice for studying plasma cytokines and brain neuroinflammation. (**B**) Radar plots representation summarizing the impact of NaCl/placebo, vitamin C or Qiseng^®^ and 5-FU/placebo, vitamin C or Qiseng^®^ treatments on plasma cytokine levels of treated mice. Cytokines in gray indicate no difference between NaCl and 5-FU conditions. Data represent the normalized mean difference between NaCl and 5-FU as percentage. NaCl/placebo (*n* = 9), 5-FU/placebo (*n* = 8–9), NaCl/vitamin C (*n* = 8–9), 5-FU/vitamin C (*n* = 6–8), NaCl/Qiseng^®^ (*n* = 10), 5-FU/Qiseng^®^ (*n* = 8–9). * *p* < 0.05, ** *p* < 0.01, *** *p* < 0.001. (**C**) Correlation between the different bacteria taxa (except domains) evaluated by qPCR and the different cytokines measured in plasma of NaCl/ and 5-FU/placebo, vitamin C and Qiseng^®^ mice. Beta-, Gamma- and Delta-proteobacteria are the significantly modified group compared with others under Qiseng^®^ treatment. Lactobacillus and some subspecies (*L. reuteri, acidophilus and johnsonii/gasseri*) corresponded to the taxa found increased after chemotherapy administration. Heatmap of Kendall correlation coefficients from medians: only significant correlations are displayed (adjusted *p* value < 0.05). The blue color indicates a positive correlation and the red color a negative correlation. The color intensity is proportional to the correlation coefficients. NaCl/placebo (*n* = 6–9), 5-FU/placebo (*n* = 8–9), NaCl/vitamin C (*n* = 8–9), 5-FU/vitamin C (*n* = 6–8), NaCl/Qiseng^®^ (*n* = 10), 5-FU/Qiseng^®^ (*n* = 8–9). 5-FU: 5-fluorouracil, L.: Lactobacillus, GMCSF: Granulocyte Macrophage Colony-Stimulating Factor, IL: Interleukin, MCP-1: Monocyte Chemotactic Protein-1, TNF-α: Tumor Necrosis Factor-α, RANTES: Regulated on Activation Normal T cell Expressed and Secreted.

**Figure 8 cancers-14-04403-f008:**
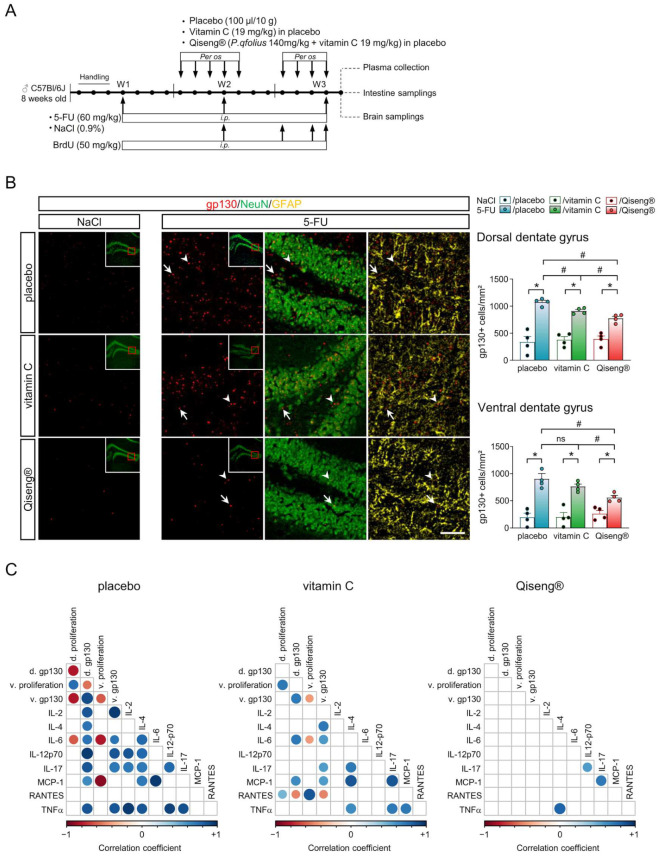
Effect of 5-FU chemotherapy treatment on brain Il-6 gp130 receptor in dorsal and ventral hippocampus. (**A**) Schematic timeline for treatments and plasma and brain sampling from the different groups of mice to investigate IL-6-related neuroinflammatory markers. (**B**) Representative microphotography of the IL-6 receptor gp130 (red) co-stained with the mature neuron NeuN (green) and astrocyte GFAP (yellow) markers in the dHp. The boxed areas show a magnification of gp130/GFAP main co-labelings (white arrows) in the dentate gyrus of the dHp. Scale bar: 100 μm. Bar plots represent mean ± SEM with symbols of individual data points of the effects of 5-FU in the presence of placebo, vitamin C and Qiseng^®^ compared with NaCl on gp130 positive cells in the dHp and vHp compared with their respective controls. Star indicates significant difference within the same groups, and hash indicates significant difference among 5-FU groups (Kruskal–Wallis-Dunn post hoc; * or # *p* < 0.05, ns: not significant). (**C**) Correlation between neural stem cell proliferation, gp130 expression in the dHp and vHp, and plasma cytokine levels. Heatmap of Kendall correlation coefficients; only significant correlations found in 5-FU mice are displayed (adjusted *p* value < 0.05). The blue color indicates a positive correlation and the red color a negative correlation. Color intensity and round size are proportional to the correlation coefficients from 0 to 1/−1. In each analysis, NaCl/placebo (*n* = 4), 5-FU/placebo (*n* = 4), NaCl/vitamin C (*n* = 4), 5-FU/vitamin C (*n* = 4), NaCl/Qiseng^®^ (*n* = 4), 5-FU/Qiseng^®^ (*n* = 4). 5-FU: 5-fluorouracil, BrdU: 5-bromo-2′-deoxyuridine, L.: Lactobacillus, GMCSF: Granulocyte Macrophage Colony-Stimulating Factor, IL: Interleukin, MCP-1: Monocyte Chemotactic Protein-1, TNF-α: Tumor Necrosis Factor-α, RANTES: Regulated on Activation Normal T cell Expressed and Secreted, GFAP: Glial fibrillary acidic protein, gp130: receptor cytokines Glycoprotein 130 kDa, NeuN: Neuronal Nuclei Antigen.

## Data Availability

Data supporting the reported results are available upon request.

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
