# Peer review of "A Panax quinquefolius-Based Preparation Prevents the Impact of 5-FU on Activity/Exploration Behaviors and Not on Cognitive Functions Mitigating Gut Microbiota and Inflammation in Mice"

_cancers, 2022, doi:10.3390/cancers14184403_

Round 1

Reviewer 1 Report

Parment and colleagues conducted a comprehensive study into the effectiveness of Panax quinquefolius (Qiseng), a ginseng-based solution, in preventing the development of chemotherapy-related cognitive impairment following systemic treatment of 5-FU in C57Bl/6j mice.They confirmed a 5-FU induced suppression of activity levels and exploratory behaviours in the open filed task and light-dark box, which were entirely prevented by co-administration of 5FU+ Qiseng (but not by coadministration of 5FU+vitamin C alone). Normal activity levels during the active phase of the mouse’s circadian cycle was also confirmed by actimetry in 5-FU+Quiseng, suggesting that Qiseng prevents chemotherapy-induced disruption to normal activity levels and natural exploratory behavior. The treatment dosage and schedule of 5-FU administration did not disrupt affective-based measures of anhedonia and anxiety-like behaviours, but was sufficient to induce spatial learning and memory deficits and poor reversal learning in the watermaze task, that were not prevented by Qiseng.Reduced rates of hippocampal neurogenesis was confirmed in 5-FU treated mice, with lower BrdU labelling relative to saline treated mice. Qiseng was largely effective in preventing chemotherapy-induced neurogenic deficits in the hippocampus. Qiseng also mitigated the chemotherapy-induced intestinal dysbiosis and systemic inflammation.

The study is well designed and straightforward. I have only relatively minor comments for the authors to address:

-       The authors refer to “activity levels/fatigue” Are these considered the same process?

-       In the 7th citation (line 85), the authors only focuses on the neurobiological underpinning of CRCI and fatigue in cancer patients. There is very little mention of mood disorders (no mentions of anxiety or depression). Further support that the neurobiological mechanisms that underlie mood disorders are pro-inflammatory cytokines may require additional sources. 

-       Justification for the use of exclusively male mice should be provided. Potential sex differences in CRCI research is a particularly important issue to consider, given the high incidence of CRCI in female breast cancer patients.

-       Male mice group housed with 8-10 males per box. Given the high incidence of aggression when C57Bl/6j male mice are group housed, can the authors address how this was assessed or controlled for? 

-       The methods section describing how the authors classified the search strategies during the MWM training should be reviewed for clarity. 

-       Specify the number of subjects for each sub-experiment for each of the 6 treatment conditions. I.e., how many mice per condition completed the behavioral tests? Were any mice excluded as outliers in behavioral testing?

-       No major differences in exploratory swimming activity (speed, distance during the visible platform trials) was observed between 5FU treated mice and saline treated mice in the MWM training, despite differences in exploratory behaviours in the open field, actimetry, etc. Can the authors speculate on this? Might chemo-induced fatigue be more apparent in voluntary exploratory behaviors like the light dark box or open field, and less apparent in aversive escape-motivated exploratory activity like the MWM?

-       I commend the authors for their use of the violin plots showing the distribution of the data.  

-       Given the significant 5FU-related weight loss in chemotherapy-treated mice, how did the authors evaluate the body condition and energy levels of the mice to ensure that any deficits in behavioral performance were not associated with sickness behavior? 

-       Figure 3 B – the panel is labelled ‘neurogenesis in the ventral dentate gyrus’, but the images of NeuN & BrdU are taken from the dorsal dentate gyrus. 

-       Figure 3B and 3C – I may have missed it, but it is not clear how the BrdU graphs shown in figure 3B and 3C differ. Are they from different sub-experimental brains? This should be made more explicit in the figure legend. Figure 3 should be revisited for clarity. 

-       While I am not an expert in the field of gut microbiota and immune responses, my understanding is that Lactobacilli is typically considered a beneficial pro-biotic in maintaining gut health. Lactobacillus is has been used in clinical studies to treat physical health disorders (Di Cerbo et al., 2016), and it can be argued that more lactobacilli in the gut microbiome can be beneficial due to its inhibition of pathogenic bacteria proliferation (Plaza-Diaz et al., 2019). But in this manuscript, increased lactobacillus count is argued to contribute to gut dysbiosis due to its role in proinflammatory pathways. Can the authors further clarify the physiological mechanisms of Lactobacilli, and why it might be deleterious in under these conditions.

-       The rationale for examining Qiseng ® for its benefits on cognitive functioning and fatigue in CRCI is well done in the introduction, yet there is not much speculation/hypotheses regarding Qiseng ® and its impact on mood disorders. It would be beneficials for the authors to discuss why Qiseng ® could be potentially therapeutic for mood disorders post-5FU by providing a few sources (if available).

-       Line 813, discusses the failure to detect compound K, it could be beneficial to discuss the importance or the reason of why compound K was not detected (ie. Sufficient time for metabolism to breakdown compound K into the smaller compound Protopanaxadiol).

-       The figure legend for Fig ‘5 I’ is missing. It appears to be combined with the description of 5H. The description of the clusters (which colors represent which condition) could be made more clear on the figure, it is quite difficult to follow.

-       Figure 8C – are the colorations provided in these matrices in 5-FU treated brains, or just NaCL treated brains given oral treatment of placebo (left), Vit C (middle), and Qiseng (right)? This should be clarified.

-       Consider adding a section on limitations to the discussion.

Minor comments for clarity:

-       Line 149 “the animals were handled daily for weight control”. It is unclear what this means? Do the authors mean the animals were handled, and weighed?

-       Line 165, what was the placebo solution made from?

-       In 2.8. Morris water maze test (MWMT), what would the researcher do when the mice failed to find the platform within the allotted 60 seconds maximum for each test? 

-       Check for typos and proof read for English grammar

o   i.e., Simple summary “Chemotherapy-related cognitive impairment (CRCI) and fatigue worsening  quality of life (QoL) of cancer patients” should say “fatigue worsens quality of life”

o   i.e., line 234 “filled with tap water at 25°C with no possibility of get out” should say “no possibility of escape”

o   Figure 3B, 3C, 8B – dentate gyrus is labelled “gyrus dentate” this should be corrected

o   line 268: “along the 4 consecutive training days and  ruing long-term memory test were obtained” It is unclear what ‘ruing long term memory’ means.

Author Response

  1. Responses to Reviewer 1

We thank the reviewer for his positive opinion on our study design, and for the number of comments and relevant questions that led to a marked improvement of the manuscript.

General comment: The manuscript was revised by following the suggestions and comments of the reviewers 1, 2 and 3. In particular, the paper was entirely corrected for English typos and grammatical errors, and also to avoid redundancies and verbose formulations. The Figure 3 was modified. Also, one co-author (Mrs Charlène Guérin) working on the qPCR detection of some gut bacteria was missing in the first submission, and was thus added in the revised manuscript.

Main modifications were highlighted in yellow. One major modification resides in a new paragraph dealing with the “limitations” at the end of the discussion section, as recommended by Reviewer 1.

Question 1: The authors refer to “activity levels/fatigue” Are these considered the same process?

We understand that the association in animals between activity levels and fatigue can be considered as extrapolation. Considering that no one has a straight answer to this question, we introduce the following paragraph in the new “limitations” section at the end of the discussion section, p35, lines 1390-1402.

“We however recognize that this work has several limits. First, we are fully aware that we interpreted “fatigue” in animals through the association of exploratory and motivation behaviors and spontaneous activity, and that we cannot claim that it is real fatigue. In patients, cancer-related fatigue (CRF) as defined by the National Comprehensive Cancer Network (NCCN) is “a distressing, persistent, subjective sense of physical, emotional and/or cognitive tiredness or exhaustion related to cancer or cancer treatment that is not proportional to recent activity and interferes with usual functioning” (Berger et al., 2015). Interestingly, it has been shown in humans that decreased motivation and fatigue are both related to alterations in the basal ganglia and reduced neuronal activation of reward signals (Miller et al., 2014 ; Karshikoff et al., 2017), suggesting similar neurological substrates. Thus, we and others focused on cancer-related fatigue in animals related to tumor growth (Dougherty et al., 2017), chemotherapy (Zombeck et al., 2013), or radiation (Renner et al., 2016), by using a common physical activity criterion also involving motivation (Perals et al., 2017).”

Question 2: In the 7th citation (line 85), the authors only focuses on the neurobiological underpinning of CRCI and fatigue in cancer patients. There is very little mention of mood disorders (no mentions of anxiety or depression). Further support that the neurobiological mechanisms that underlie mood disorders are pro-inflammatory cytokines may require additional sources. 

We agree with the reviewer that this remains an important question, even if until know the causal links between cancer and treatments on anxiety and depressive disorders are not clearly established in patients. To meet this important question, we add the following sentences in the introduction section and the references dealing with cytokine levels and mood disorders, page 2, lines 88-93.

In patients with various cancers in whom prevalence of depressive and anxiety symptoms were around 23 and 19 % respectively (Naser et al., 2021), there is however a lack of association between cytokine levels and mood, most of them receiving at least chemotherapy. Of note, some meta-analyses established that cytokines including IL-6, IL-10, TNF-α and MCP-1 were found increased in patients with depression compared with healthy controls (Himmerich et al., 2019, Köhler et al. 2018).”

Question 3: Justification for the use of exclusively male mice should be provided. Potential sex differences in CRCI research is a particularly important issue to consider, given the high incidence of CRCI in female breast cancer patients.

The thank the reviewer who raises this important question: most clinical studies on CRCI indeed concern female breast cancer population. Given the paucity of publications in male patient cohorts, we can ask whether cancer men and women treated with chemotherapy were affected differently by CRCI. We agree that our animal models should take into account this gender diversity. we add a discussion part on the limitations. Page 35, lines 1402-1411.

“Second, this work was exclusively done on C57Bl/6 male mice to avoid behavioral issues due the estrous cycle in female mice. Accordingly, it was shown in female mice that the estrus cycle has a robust effect on anxiety, low anxiety being associated with in high-estradiol level (proestrus and estrus phase) (Wang et al., 2019). Nevertheless, a very important study in young mice exposed to a combination of chemotherapy, established that males and not females exhibit deficits in short memory and executive functions 4-5 weeks after chemotherapy (Konsman et al., 2021). Interestingly, this early exposure to chemotherapy led to an increased MCP-1 expression only in males, while oligodencrocyte markers were altered in both males and females (Konsman et al., 2021). Thus, the present work should be reproduced in female mice.”

Question 4: Male mice group housed with 8-10 males per box. Given the high incidence of aggression when C57Bl/6j male mice are group housed, can the authors address how this was assessed or controlled for? 

In agreement with the ethics committee, the animals were housed in groups in regulation-sized cages with enrichment (Lignocel®, JRS, Rosenberg, Germany) in order to keep the mice busy and limit fighting. Animals were provided by an authorized supplier "Janvier Labs" and the groups per cage were kept on reception, in order to preserve the hierarchy established within the group. To reduce the stress, all animals were familiarized with handlings and weightings daily before experiments. Thus, mice were daily monitored to detect any suffering (weight loss, "hunched back" posture, slowness of movement, prostration, large wounds fights, bristling hair, etc.). This monitoring allowed us to quickly detect any sign of suffering and to put an end to it. In spite of some aggressiveness of some animals and observed fights, the injuries remained limited, never impacting the animal's health.

Question 5: The methods section describing how the authors classified the search strategies during the MWM training should be reviewed for clarity. 

According to the recommendation of the reviewer, the MWM automated detection of learning strategies was rewrote and reduced for clarity, one important reference (Nicola et al., 2021) was added, in the Material and Method section: page 6, lines 277-297.

Question 6: Specify the number of subjects for each sub-experiment for each of the 6 treatment conditions. I.e., how many mice per condition completed the behavioral tests? Were any mice excluded as outliers in behavioral testing?

The number of animals per analysis and per group was specified in each legend (highlighted in Yellow). There was no evidence of outliers on the behavioral data.

Question 7: No major differences in exploratory swimming activity (speed, distance during the visible platform trials) was observed between 5FU treated mice and saline treated mice in the MWM training, despite differences in exploratory behaviours in the open field, actimetry, etc. Can the authors speculate on this? Might chemo-induced fatigue be more apparent in voluntary exploratory behaviors like the light dark box or open field, and less apparent in aversive escape-motivated exploratory activity like the MWM?

This is an interesting question. In our study, this is during the familiarization day (D8) when the platform is visible, that we did not detect a difference between the different groups e.g., 5-FU and NaCl groups, in the platform location latency, distance travelled and swimming speed. The sentence, page 14 and line 629, was rewritten for better clarity. This observation suggests that 5-FU did not drastically impact physical, muscular or visual capacities in treated animals. However, during the training phase of MWMT, 5-FU mice showed higher latency to find the non-visible platform compared to controls, indicative of spatial learning alteration, a cognitive process non-compensated by Qiseng®. Interestingly, the voluntary exploratory behavior also involving aversive environment related to fatigue (in LDB test or OFT), impacted by 5-FU, was prevented by Qiseng®. There fatigue and cognitive deficits induced by 5-FU may be sustained by the neurogenesis defects in ventral and ventral hippocampus, respectively. We think that the fatigue symptom is connected to voluntary activity but also motivation to exploration behaviors, that probably explains alteration in the exploratory learning strategies (cognitive scores in MWMT).

Question 8: Given the significant 5FU-related weight loss in chemotherapy-treated mice, how did the authors evaluate the body condition and energy levels of the mice to ensure that any deficits in behavioral performance were not associated with sickness behavior? 

Animals were monitored daily throughout the treatment period and during behavioral testing for signs of suffering. When an animal showed significant weight loss (more than 15% of the pre-treatment weight), a "hunched back" posture, slowness of movement, prostration, significant wounds (fights) or bristly hair, we considered that a limit of suffering had been reached and the animal was sacrificed by decapitation under anesthesia. For the very large majority of animals, their weight recovered during the W4 after the last 5-FU injection before starting behavior evaluations. We think that i) the normal locomotion of 5-FU/Qisengâ mice in OFT while they exhibit the same weight loss than 5-FU/NaCl, ii) the same behavior of 5-FU and NaCl mice in the FST and TST, and iii) the absence of 5-FU effect on swimming capacity at D8 in the MWMT, suggest that sickness is not involved in behavioral performance defects induced by 5-FU.

Question 9: Figure 3 B – the panel is labelled ‘neurogenesis in the ventral dentate gyrus’, but the images of NeuN & BrdU are taken from the dorsal dentate gyrus. 

Thank you for your alertness. We apologize for this mistake. The label in panel 3B has been corrected into "Neurogenesis in the dorsal dentate gyrus".

Question 10: Figure 3B and 3C – I may have missed it, but it is not clear how the BrdU graphs shown in figure 3B and 3C differ. Are they from different sub-experimental brains? This should be made more explicit in the figure legend. Figure 3 should be revisited for clarity. 

As recommended, the Figure 3 has been revised for clarity, and the two quantification panel graphs were indeed from sub-experimental brains and were relabeled “Dorsal dentate gyrus” and Ventral dentate gyrus, in Figure 3B.

Question 11: While I am not an expert in the field of gut microbiota and immune responses, my understanding is that Lactobacilli is typically considered a beneficial pro-biotic in maintaining gut health. Lactobacillus is has been used in clinical studies to treat physical health disorders (Di Cerbo et al., 2016), and it can be argued that more lactobacilli in the gut microbiome can be beneficial due to its inhibition of pathogenic bacteria proliferation (Plaza-Diaz et al., 2019). But in this manuscript, increased lactobacillus count is argued to contribute to gut dysbiosis due to its role in proinflammatory pathways. Can the authors further clarify the physiological mechanisms of Lactobacilli, and why it might be deleterious in under these conditions.

This is a very interesting question. The Lactobacillus genera include a wide diversity of species that have versatile effects on host physiology. These bacteria have been widely used as probiotics to improve gastrointestinal health. Indeed, these bacteria were reported to promote mucus and anti-microbial peptides, to decrease intestinal permeability and to provide colonization resistance against pathogens (Dempsey et al., Frontiers in Immunology, 2022). However, increased levels of Lactobacilli are not always associated with healthy conditions. For example, higher levels of Lactobacilli were reported in obese individuals (Million et al., International Journal of Obesity, 2013) or in mouse models of undernutrition (Breton et al., Clin Nutr, 2021). In the case of inflammation, different species of Lactobacilli were reported to trigger immunotolerance while others promote inflammation (Dempsey et al., Frontiers in Immunology, 2022). For instance, high proportion of Lactobacillus in gut was positively correlated with inflammatory diseases such as the Crohn's disease (Wang et al., 2014). Some examples of the mechanisms triggered by Lactobacilli that induce inflammation are now discussed in the manuscript. Page 33, lines 1289-1295.

“Although several Lactobacillus species are currently used as probiotics to improve gastrointestinal health, increased levels of Lactobacilli are not always associated with healthy conditions. For example, higher levels of Lactobacilli were reported in obese individuals (Million et al., International Journal of Obesity, 2013), as well as in mouse models of undernutrition (Breton et al., Clin Nutr, 2021). Lactobacilli may in some instances activate the immune system, by engaging host pattern-recognition receptors, and trigger inflammation (Dempsey et al., Frontiers in Immunology, 2022) playing a role in bowel inflammation disease (Wang et al., 2014)».

Question 12: The rationale for examining Qiseng ® for its benefits on cognitive functioning and fatigue in CRCI is well done in the introduction, yet there is not much speculation/hypotheses regarding Qiseng ® and its impact on mood disorders. It would be beneficials for the authors to discuss why Qiseng ® could be potentially therapeutic for mood disorders post-5FU by providing a few sources (if available).

We admit that depression or anxiety is diagnosed in around 20% of cancer patients. We agree with the reviewer that this question should be of interest, but our mouse study fails to demonstrate a 5-FU chemotherapy-induced anxiety or depression. To answer the reviewer question, it was shown that mice under sleep deprivation exhibit a significant decrease in locomotor activity and in anxiety-like behavior, and that Panax quinquefolius for 8 days significantly improved locomotor activity and anxiety state (Chanana et al. 2016). Other preclinical studies indicate that some high doses of ginsenosides can improve anxiety or depressive state in animal models. But in the absence of mood disorders in 5-FU mice, we think that discussing the role of ginsenosides on mood will risk to add complexity and speculation.

Question 13: Line 813, discusses the failure to detect compound K, it could be beneficial to discuss the importance or the reason of why compound K was not detected (ie. Sufficient time for metabolism to breakdown compound K into the smaller compound Protopanaxadiol).

We agree that we should try explaining the absence of detection of compound K in mouse sera. Theses sentences were added to complete this part in the discussion section, page 33, lines 1306-1312.

“The absence of compound K can be explained by a rapid metabolization. Indeed, in rats, it was shown that compound K was detectable in blood from 4h to 7h post-oral administration of Rb1 (Akao et al., 1998). Since we collect blood samples at least 24h post Qiseng® oral treatment, the absence of compound K can be correlated to the its metabolization into the Protopanaxadiol, here detected in some samples. The levels of ginsenosides found nevertheless after 24h, are in agreement with previous studies showing a proportion of Rb1 and Rc ginsenosides unmetabolized by bacteria after their administration [90,91].

Question 14: The figure legend for Fig ‘5 I’ is missing. It appears to be combined with the description of 5H. The description of the clusters (which colors represent which condition) could be made more clear on the figure, it is quite difficult to follow.

We agree with the redundancy and lack of clarity. Thus we proposed in the revision only one representation of the PCA in a single panel H, and we modified symbols to help for clarity in the reading of the different groups which composed cluster 1, cluster 2 and cluster 3. The Figure legend was modified accordingly, page 33, lines 947-948.

Question 15: Figure 8C – are the colorations provided in these matrices in 5-FU treated brains, or just NaCL treated brains given oral treatment of placebo (left), Vit C (middle), and Qiseng (right)? This should be clarified.

We made the correction by precising the attribution of colorations to placebo, vitamin C and Qiseng® in the Figure 8 Legend. Page 30, line 1115.

Question 16: Consider adding a section on limitations to the discussion.

We agree with the reviewer that this should be a plus in the present paper and we add this “limitations” part in the discussion section. We included the discussion on animal sex and on the relationship between “fatigue” and activity/exploration.

Page 33, lines 1390-1411.

Minor comments for clarity:

-Question 1: Line 149 “the animals were handled daily for weight control”. It is unclear what this means? Do the authors mean the animals were handled, and weighed?

The animals were handled, and daily weighted. The sentence was corrected for clarity. Page 4, lines 160-161.

-Question 2: Line 165, what was the placebo solution made from?

The placebo solution contains: Vegetable glycerin (stabilizer), Stevia sweetener, Raspberry flavor, i.e., the solution contained all ingredient of the formulated product but the active ingredients, vitamin C and the Panax quinquefolius extract. This composition was added in the Material and Methods section, page 4, lines 178-179.

-Question 3: In 2.8. Morris water maze test (MWMT), what would the researcher do when the mice failed to find the platform within the allotted 60 seconds maximum for each test? 

If the animal could not find the location of the platform after 60 seconds, the mouse was gently positioned on it for about 20-30 seconds to allow spatial localization and symbols detection, before being removed from the basin.

-Typos and proof read for English grammar have been checked

  1. Simple summary “Chemotherapy-related cognitive impairment (CRCI) and fatigue worsening  quality of life (QoL) of cancer patients” should say “fatigue worsens quality of life”
  2. e., line 234 “filled with tap water at 25°C with no possibility of get out” should say “no possibility of escape”
  3. Figure 3B, 3C, 8B – dentate gyrus is labelled “gyrus dentate” this should be corrected
  4. line 268: “along the 4 consecutive training days and  ruing long-term memory test were obtained” It is unclear what ‘ruing long term memory’ means.

We apologize, the number of typos and grammar errors has been corrected.

I would like to thank you for your constructive remarks.

Sincerely yours,

Dr Hélène Castel,

Inserm Research Director, U1245, 25 rue Tesnière, University of Rouen Normandie, 76821 Mont-Saint-Aignan

Reviewer 2 Report

Parment et al. reported that Qiseng@ prevented the reduction in activity/exploration and symptoms of fatigue induced by 5-FU, and dampens chemotherapy induced intestinal dysbiosis, systemic inflammation, neuroinflammation, and neurogenesis damage. It provides useful information for CRCI prevention. Major concerns include:

1.       They detected the serum ginsenosides in mice receiving Qiseng@. Did they detect the brain ginsenosides in mice receiving Qiseng@? And were the ginsenosides in serum/ brain of mice receiving Qiseng@ associated with the intestinal dysbiosis, systemic inflammation, neuroinflammation, and neurogenesis damage?

2.       About the discussion of the mechanism of Qiseng@, it is better to condense the discussion of the mechanism into one paragraph to improve the readability.

3.       It remains elusive why Qiseng@ dampens chemotherapy induced intestinal dysbiosis. Please discuss.

Author Response

Responses to Reviewer 2:

We would like to thank the reviewer for his interest and the important questions he raised.

General comment: The manuscript was revised by following the suggestions and comments of the reviewers 1, 2 and 3. In particular, the paper was entirely corrected for English typos and grammatical errors, and also to avoid redundancies and verbose formulations. The Figure 3 was modified. Also, one co-author (Mrs Charlène Guérin) working on the qPCR detection of some gut bacteria was missing in the first submission, and was thus added in the revised manuscript.

Main modifications were highlighted in yellow. One major modification resides in a new paragraph dealing with the “limitations” at the end of the discussion section, as recommended by Reviewer 1.

Question 1: They detected the serum ginsenosides in mice receiving Qiseng@. Did they detect the brain ginsenosides in mice receiving Qiseng@? And were the ginsenosides in serum/brain of mice receiving Qiseng@ associated with the intestinal dysbiosis, systemic inflammation, neuroinflammation, and neurogenesis damage?

We agree that these experiments would have provided key information about potential direct effects of ginsenosides within the brain. Indeed, it has been shown in rats, that Rb1 ginsenosides could be found in the brain up to 24 hours intravenous post-injection (Wang et al., 2018). We did not attempt brain distribution, but regarding the faint levels of ginsenosides found in mouse sera, we think that in our experimental conditions, it would have been difficult to detect some ginsenosides in brain samples.

No correlation study was performed between the proportions of ginsenosides in the brain or the serum, and the other inflammatory and neurobiological disorders, due to the inter-individual variability of the ginsenosides types in chemotherapy and non-chemotherapy-treated mice. But according to the literature, ginsenosides can be associated with the dysbiosis observed in 5-FU and NaCl Qiseng® groups. Indeed, a human study on the metabolism of Panax ginseng showed that low fecal levels of Proteobacteria was associated with high metabolism of Rb1 ginsenosides to compound K (Song et al., 2013). We can just hypothesize that decrease in the proportion of Proteobacteria observed in the Qiseng® group results from Rb1 metabolism.

Question 2: About the discussion of the mechanism of Qiseng@, it is better to condense the discussion of the mechanism into one paragraph to improve the readability.

We now condense the discussion on Qiseng in a single paragraph.

Question 3: It remains elusive why Qiseng@ dampens chemotherapy induced intestinal dysbiosis. Please discuss.

We agree with the reviewer but we already tried in the first version of the manuscript to attempt discussing without being too speculative.

Studies show that chemotherapy treatment induces intestinal dysbiosis leading to gut damage through toll-like receptor (TLR) signaling pathways (Wei et al., 2021) and/or the generation of cellular reactive oxygen species (ROS) (Sonis et al., 2004). In our study, we also observed that 5-FU provokes dysbiosis, intestinal damage, macrophage infiltration, systemic inflammation. The following sentence was added in the discussion section, pages 1280-1285.

“Thus, Qiseng® modifies gut microbiota equilibrium through a decrease of proteobacteria, while preventing the increase of Lactobacillus induced by 5-FU. Similarly, ginsenoside supplementation can restore the composition of the microbiota altered by chemotherapy (Zhou et al., 2021). Also, it was shown that a decoction of Ginseng could restore D-galactose-induced memory deficits in rats by up-regulating Bacteroidetes and down-regulating Lactobacillus (Wang et al., 2021).”

I would like to thank you for your constructive remarks.

Sincerely yours,

Dr Hélène Castel,

Inserm Research Director, U1245, 25 rue Tesnière, University of Rouen Normandie, 76821 Mont-Saint-Aignan

Reviewer 3 Report

Parment et al. have provided a tour de force manuscript exploring the therapeutic potential of Qiseng and vitamin c on 5-FU chemotherapy-induced neurotoxicities. This work is timely and would be of interest to the readers of Cancers. I have a number of questions/issues that should be addressed before acceptance for publication. 

Major:

1.     Why was the dose of 5FU, Vit C and Qiseng chosen? This is important due to the mortality rate (Fig1C). Was the dose too high and therefore unlikely for the treatments to have a chance of remediating some of the 5FU-induced cognitive side-effects? Was the mortality rate unexpected? Were animals euthanased because they had reached ethical requirements for euthanasia (and if so, what were they) or found dead?

2.     Related to the comment above, little time is given in discussing the mortality findings, which could have major clinical implications; namely do the findings suggest that Vitamin C treatment may be unsafe for patients receiving chemotherapy?

3.     At times, the focus of analysis and interpretation seems to be the comparison between NaCl/Qiseng vs 5FU/Qiseng groups (e.g. Fig 3C). This is a curious approach given that this implies the research question is whether 5FU impacts the effects of Qiseng. Post hoc analyses that compare between 5FU/placebo and 5FU/Qiseng would be a more relavent approach to determining if Qiseng improvise deficits caused by 5FU.

4.     The manuscript is thorough but very verbose with methods provided in both the Methods and Results. A more succinct approach may help achieve a wider audience which this work deserves.

5.     I would have liked to have read about some of the potential clinical implications in the discussion.

Minor:

1.     Title: the use of “via” in the title may be misleading. “and” may be better given the data show that microbiota and inflammation changes are associated with Qiseng’s effects on activity but not necessarily causation.

2.     Here CRCI refers to chemotherapy-induced cognitive impairment whereas CRCI is more widely referred to as cancer-related cognitive impairment. This may cause confusion.

3.     There are some minor grammatical issues throughout. A thorough read over would help. Some examples are provided below:

Simple summary: the opening sentence is more of a heading than a sentence.

 Pg1 line 35: “macrophages” should be “macrophage”

Pg4, line 171: “weighed” not “weighted”

Page 27, line 990: “cytokine” not “cytokines”

Pg30, line 1069: should be “implemented a mouse model”

Pg30, line 1075: should this be “5FU-related”?

Pg31, line1100: what are “strong” criteria?

Pg31, line1107: grammatical issue - “they less explore”

Pg31, line1101: when discuss rodent behavior “depression-like” should be used

Pg34, line1274: The sentence starting with “here Qiseng administered…” has several grammatical issues that make it hard to understand. Suggest rewriting this. further, how does Qiseng block serum Il6? Do you mean “induction of Il6”?

Author Response

Responses to Reviewer 3:

We thank the reviewer for his important remarks and comments.

General comment: The manuscript was revised by following the suggestions and comments of the reviewers 1, 2 and 3. In particular, the paper was entirely corrected for English typos and grammatical errors, and also to avoid redundancies and verbose formulations. The Figure 3 was modified. Also, one co-author (Mrs Charlène Guérin) working on the qPCR detection of some gut bacteria was missing in the first submission, and was thus added in the revised manuscript.

Main modifications were highlighted in yellow. One major modification resides in a new paragraph dealing with the “limitations” at the end of the discussion section, as recommended by Reviewer 1.

Question 1: Why was the dose of 5FU, Vit C and Qiseng chosen? This is important due to the mortality rate (Fig1C). Was the dose too high and therefore unlikely for the treatments to have a chance of remediating some of the 5FU-induced cognitive side-effects? Was the mortality rate unexpected? Were animals euthanased because they had reached ethical requirements for euthanasia (and if so, what were they) or found dead? Question 2: Related to the comment above, little time is given in discussing the mortality findings, which could have major clinical implications; namely do the findings suggest that Vitamin C treatment may be unsafe for patients receiving chemotherapy?

The dose of 5-FU used in this study was 60 mg/kg once a week for 3 weeks. Previous studies indicate that administration of 5-FU (50 or 100 mg/kg per day, for five days) to mice was found to result in low mortality, but was also the optimal dose to induce intestinal mucositis in mice (Zhang et al., 2018). Other studies have used this same 60 mg/kg dose (once daily for 5 consecutive days) in mice without reporting mortality (Dougherty et al., 2019 ; Hsu et al., 2020).

 “The 60 mg/kg dose was optimal to induce intestinal mucositis in mice (Zhang et al., 2018) without reporting mortality (Dougherty et al., 2019 ; Hsu et al., 2020). It can be hypothesized that the deadly effect of chemotherapy combined to stress (handling, i.p. injections combined to gavage) and the daily gavage by itself potentially responsible for inflammatory lesions of the esophagus, may have contributed to the observed mortality.”

We don’t consider this statistical since this data was obtained for the experimental protocol of Figure 1A including two sequential series of experiments including 24-26 mice in each the placebo/5-FU, vitamin C/5-FU and Qiseng®/5-FU groups, leading to 3, 4 and 1 death, respectively due to the treatments. The hypothesis on mortality was proposed in the discussion section 31, lines 1170-1175.

Concerning dosages of Qiseng® and vitamin C: either an oral LD50 of 750 mg ginseng/kg for rats and 200 mg/kg for mice was reported in the literature (RTECS, 1998) or no toxicity at all (From National Toxicology Program publication. Toxicology and carcinogenesis studies of ginseng (CAS No. 50647-08-0) in F344/N rats and B6C3F1 mice (gavage studies). Natl Toxicol Program Tech Rep Ser. 2011 Sep;(567):1-149.) was found. Based on that, we made the choice to test the daily dose recommended by the supplier, e.g. 30 ml containing 430.1 mg de P.qfolius; each mouse (around 20 µg) received 200 µl per os of Qiseng® solution containing 2.87 mg de P.qfolius (140 mg/kg).

Considering that Qiseng® solution contained P. qfolius (25% ginsenosides) and vitamin C, corresponding to 19 mg/kg in a solution of 200 µl, we prepared an identical daily concentration of vitamin C (19 mg/kg) in the placebo solution. Administration of 100 or 200 mg/kg of P.qfolius for 8 days (Chanana et al., 2016) or vitamin C at 100 or 200 mg/kg for 10 days (Hasan Khudhair et al., 2022) did not induce mouse death.

Animals were monitored daily throughout the treatment period and during behavioral testings for signs of suffering. When the animals showed significant weight loss (more than 15% of the pre-treatment weight), a "hunched back" posture, slowness of movement, prostration, significant wounds (fights) or bristly hair, we considered that a limit of suffering had been reached and animals were sacrificed by decapitation under anaesthesia by isoflurane inhalation (5%). Of all the animals in the study, only 3 were found dead the day after treatment with no visible signs of suffering or sickness before and after treatment. For the others included in the calculation of the mortality rate, they were sacrificed following the treatments, because they showed a significant weight loss, a stooped posture and bristly hairs.

Question 3: At times, the focus of analysis and interpretation seems to be the comparison between NaCl/Qiseng vs 5FU/Qiseng groups (e.g. Fig 3C). This is a curious approach given that this implies the research question is whether 5FU impacts the effects of Qiseng. Post hoc analyses that compare between 5FU/placebo and 5FU/Qiseng would be a more relavent approach to determining if Qiseng improvise deficits caused by 5FU.

We apologize and added post-hoc results on histograms of the panels 3B and 3C. In Figure 3B and 3C, the difference between 5-FU/placebo and 5-FU/Qiseng® is close to significance (P value of P=0.06) and significant, respectively.

Question 4: The manuscript is thorough but very verbose with methods provided in both the Methods and Results. A more succinct approach may help achieve a wider audience which this work deserves.

We agree and made efforts to rewrite the entire manuscript in order to be more concise and less verbose.

Question 5: I would have liked to have read about some of the potential clinical implications in the discussion.

A paragraph was added in the conclusion section to convince on the clinical impact of our preclinical research in fatigue induced by chemotherapy, pages 34 and 35, lines 1433-1444.

“This data provide an important validation of the beneficial effect of a Qiseng® supplementation on fatigue in cancer patients treated by chemotherapy, while this type of non-pharmacological intervention should be more accepted by patients. Accordingly, a recent clinical trial “Evaluation of the Impact of Taking American Ginseng for 8 Weeks on Fatigue in Patients Treated for Localized Breast Cancer (QISEIN)” (NCT05241405) was initiated very recently with the objective of improving cancer patients QoL. We think that Qiseng®-induced benefit is not trivial, since mental fatigue affects large-scale network connectivity in the brain during high cognitive demands associated with effort. In fact, preventing neurofatigue should allow optimal cognitive functions or acceptance of the effort required in a cognitive rehabilitation protocol.

Minor Comments

Question 1: Title: the use of “via” in the title may be misleading. “and” may be better given the data show that microbiota and inflammation changes are associated with Qiseng’s effects on activity but not necessarily causation.

We made a proposition in the title to be as shorter as possible and to fulfil the reviewer request. We propose “A Panax quinquefolius-based preparation prevents the impact of 5-FU on activity/exploration behaviors and not on cognitive functions mitigating gut microbiota and inflammation in mice

Question 2: Here CRCI refers to chemotherapy-induced cognitive impairment whereas CRCI is more widely referred to as cancer-related cognitive impairment. This may cause confusion. Some sentences were adjusted to take into account this important remark.

Question 3: There are some minor grammatical issues throughout. A thorough read over would help. Some examples are provided below:

 Typos and grammatical errors were now corrected according to the remarks of the reviewer.

Simple summary: the opening sentence is more of a heading than a sentence.

We apologize, this was corrected.

Pg31, line1100: what are “strong” criteria?

The term “strong” was removed, please see the added “limitations section” for additional discussion for fatigue in animals

I would like to thank you for your constructive comments.

Sincerely yours,

Dr Hélène Castel, Inserm Research Director, U1239, 25 rue Tesnière, University of Rouen Normandie, 76821 Mont-Saint-Aignan